# Molecule Design by Latent Prompt Transformer

**Deqian Kong**[1,⋆]**, Yuhao Huang**[2,⋆]**, Jianwen Xie**[3,4,⋆]**, Edouardo Honig**[1,⋆]**,**
**Ming Xu**[5]**, Shuanghong Xue**[5]**, Pei Lin**[6]**, Sanping Zhou**[2]**,**
**Sheng Zhong**[5,6]**, Nanning Zheng**[2]**, Ying Nian Wu**[1]

[1]Department of Statistics and Data Science, UCLA
[2]Institute of Artificial Intelligence and Robotics, Xi'an Jiaotong University
[3]BioMap Research [4]Akool Research
[5]Institute of Engineering in Medicine, UCSD
[6]Shu Chien-Gene Lay Department of Bioengineering, UCSD

## Abstract

This work explores the challenging problem of molecule design by framing it as a conditional generative modeling task, where target biological properties or desired chemical constraints serve as conditioning variables. We propose the Latent Prompt Transformer (LPT), a novel generative model comprising three components: (1) a latent vector with a learnable prior distribution modeled by a neural transformation of Gaussian white noise; (2) a molecule generation model based on a causal Transformer, which uses the latent vector as a prompt; and (3) a property prediction model that predicts a molecule's target properties and/or constraint values using the latent prompt. LPT can be learned by maximum likelihood estimation on molecule-property pairs. During property optimization, the latent prompt is inferred from target properties and constraints through posterior sampling and then used to guide the autoregressive molecule generation. After initial training on existing molecules and their properties, we adopt an online learning algorithm to progressively shift the model distribution towards regions that support desired target properties. Experiments demonstrate that LPT not only effectively discovers useful molecules across single-objective, multi-objective, and structure-constrained optimization tasks, but also exhibits strong sample efficiency.

## 1 Introduction

Molecule design plays a pivotal role in drug discovery, focusing on identifying or creating molecules with desired pharmacological or chemical properties. However, the vast and complex space of potential drug-like molecules presents significant challenges for systematically designing efficient machine learning models to navigate this space. Latent space optimization (LSO) (Maus et al., 2022; Tripp et al., 2020; Gómez-Bombarelli et al., 2018), a two-stage procedure, has emerged as a promising approach to address this challenge. The first stage involves training a deep generative model, typically a deep variational auto-encoder (VAE) (Kingma and Welling, 2014), to map low-dimensional continuous vectors to the data manifold in input space, creating a simplified and continuous analog of the original optimization problem. In the second stage, the objective function is optimized over this learned latent space using a surrogate model. While LSO has demonstrated potential in tackling high-dimensional and structured input spaces, existing LSO-based methods separate the training of the generative model from the property-conditioned optimization. This decoupling can lead to suboptimal performance, as the learned latent space may not be ideally suited to the specific

---

⋆Equal Contribution. Project page: https://sites.google.com/view/latent-prompt-transformer.

optimization task. Furthermore, LSO depends on an additional inference model during training, increasing the number of model parameters and adding complexity to the model design process.

To address the above limitations, we propose a novel generative model, the Latent Prompt Transformer (LPT), which unifies molecule generation and optimization within a single framework. The LPT can be directly trained on observed molecule-property pairs via maximum likelihood estimation (MLE) in an end-to-end fashion. We take advantage of the immediately-available Markov chain Monte Carlo (MCMC) inference engine derived from the posterior distribution, eliminating the need for an auxiliary network for variational inference. The LPT consists of three components: (1) a learnable prior model of the latent vector based on a neural transformation of Gaussian noise, (2) a molecule generation model that generates molecule sequences using a causal transformer with the latent vector serving as the prompt, and (3) a property predictor model that estimates the target property values given the latent vector. The latent prompts serve as shared representations for both molecules and properties. By incorporating the MLE of the LPT, we employ an online learning algorithm of the LPT for property optimization, formulated as a conditional generative modeling task. This algorithm progressively shifts the model distribution towards regions associated with desired properties.

To enhance the practical application of designed molecules and enable a comparative analysis of our generated designs versus those made by human experts, we introduce a new task, which is the conditional generation of molecules that bind to the NAD binding site of Phosphoglycerate dehydrogenase (PHGDH). In addition to single-objective optimization for this specific protein, we also introduce structure-constrained optimization to analyze the design pathways of learning-based models in comparison with human experts. This additional experiment aims to provide valuable insights for improving learning-based model design. We manually process the data to ensure compatibility with docking software, which is essential for facilitating precise score computation. Compared to other widely used evaluation metrics, the use of PHGDH offers greater practical significance and enables more accurate calculations in real-world drug discovery scenarios.

Our work makes the following key contributions: (1) We propose the LPT, a novel generative framework for jointly modeling molecule sequences and their target properties. This framework leverages a learnable informative prior distribution on a latent space, conceptualized as an intrinsic design representation. (2) We propose an MCMC-based MLE to train the LPT without needing an auxiliary network for variational inference. (3) We develop a novel online learning algorithm for LPT, allowing the model to gradually extrapolate to feasible regions associated with desired properties, improving computational and sample efficiency. (4) We present a new task for the conditional generation of molecules that bind to the NAD binding site of PHGDH, and introduce structure-constrained optimization to compare the design pathways of learning-based models with those of human experts. (5) Our model achieves state-of-the-art performance across a variety of molecule-based optimization tasks, including single-objective design, multi-objective design, and biological sequence design.

## 2 Background

Let $x = (x^{(1)}, ..., x^{(t)}, ..., x^{(T)}) \in \mathcal{X}$ be a sequence representation of molecules such as SMILES (Weininger, 1988) or SELFIES (Krenn et al., 2020; Cheng et al., 2023), where $x^{(t)} \in \mathcal{V}$ is the $t$-th element in the vocabulary $\mathcal{V}$, and $\mathcal{X}$ is the space of molecules. In molecule design, the objective is to generate targeted molecules that optimize several properties of interest, represented by $\boldsymbol{y} = \{y_i = o_i^p(x) \in \mathbb{R}\}$ for $i = 1, \ldots, m$, while satisfying specific constraints, $\boldsymbol{c} = \{c_j = o_j^c(x) \in \mathbb{Z}_2\}$ for $j = 1, \ldots, n$. Here, $o^p(x)$ and $o^c(x)$ denote oracle functions determining property values and constraint satisfaction, respectively, and these values can be obtained either by querying existing software or by conducting wet lab experiments.

The multi-objective multi-constraint optimization (MMO) problem can be formulated as follows:
$$\max_{x \in \mathcal{X}} \quad F(x) = \{o_1^p(x), o_2^p(x), \ldots, o_m^p(x)\} \quad \text{s.t.} \quad o_j^c(x) = 1, \quad j = 1, \ldots, n, \qquad (1)$$

where $o^c(x) = 1$ denotes constraint satisfaction and $o^c(x) = 0$ denotes constraint violation. From a probabilistic modeling perspective, the MMO problem can be viewed as a conditional sampling problem, $x^* \sim p(x|\boldsymbol{y} = \boldsymbol{y}^*, \boldsymbol{c} = \boldsymbol{1})$, where $\boldsymbol{y}^*$ represents the desired values of the target properties, and $\boldsymbol{c} = \boldsymbol{1}$ indicates that all constraints are satisfied.

In generative molecule design, we assume access to a dataset of molecule sequences and their associated properties for offline pretraining of the generative model. During the design phase,

the model queries existing software (oracle functions) to obtain single or multiple metrics for the generated molecules, with these oracle functions providing ground-truth property values to be optimized. It is important to acknowledge that developing proper software for accurate evaluation of molecule properties is of equal or even greater importance than designing molecules based on those software. However, it is beyond the scope of this paper to address the development of such software. Here, we treat the property values obtained from oracle functions as ground-truths for optimization.

# 3   Latent Prompt Transformer

## 3.1   Model

Suppose $x = (x^{(1)}, ..., x^{(t)}, ..., x^{(T)})$ is a molecule sequence, (e.g. a string-based representation like SMILES (Weininger, 1988) or SELFIES (Krenn et al., 2020; Cheng et al., 2023)), where $x^{(t)} \in \mathcal{V}$ is the $t$-th element of sequence in the vocabulary $\mathcal{V}$. The latent vector is $z \in \mathbb{R}^d$ and $y \in \mathbb{R}$ is the target property value, or $y \in \{0, 1\}$ indicates constraint satisfaction. The term *property* refers to both real-valued properties and binary-valued constraints in the following sections. The joint distribution of a molecule and its property is defined as $p(x, y)$.

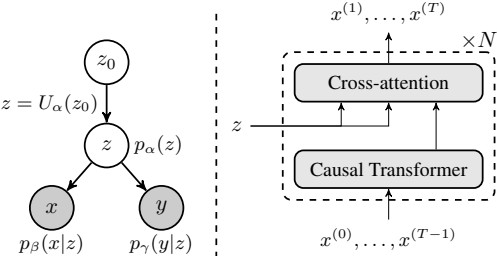

Figure 1: *Left*: Overview of Latent Prompt Transformer (LPT). The latent vector $z \in \mathbb{R}^d$ is a neural transformation of $z_0$, i.e., $z = U_\alpha(z_0)$, where $z_0 \sim \mathcal{N}(0, I_d)$. Given $z$, $x$ and $y$ are independent. $p_\beta(x|z)$ is the molecule generation model and $p_\gamma(y|z)$ predicts the property value or constraint based on $z$. *Right*: Illustration of molecule generation model $p_\beta(x|z)$. The latent vector $z$ is used as a prompt in the $p_\beta(x|z)$ via cross-attention.

The latent variable $z$, conceptualized as a design representation, decouples the molecule generation from property prediction. Specifically, given $z$, we assume $x$ and $y$ are conditionally independent, making $z$ the information bottleneck. With this assumption, our Latent Prompt Transformer (LPT) is defined as,

$$p_\theta(x, y, z) = p_\alpha(z)p_\beta(x|z)p_\gamma(y|z), \tag{2}$$

where $\theta = (\alpha, \beta, \gamma)$. $p_\alpha(z)$ is a prior model with parameters $\alpha$. Here, $z$ serves as the latent prompt for the generation model $p_\beta(x|z)$ parameterized by a causal Transformer with parameters $\beta$. $p_\gamma(y|z)$ is the predictor model with parameters $\gamma$. As shown in Fig. 1, LPT defines the generation process as,

$$z \sim p_\alpha(z), [x|z] \sim p_\beta(x|z), [y|z] \sim p_\gamma(y|z). \tag{3}$$

For the prior model, $p_\alpha(z)$ is formulated as a learnable neural transformation from an uninformative distribution, such as an isotropic Gaussian, $z = U_\alpha(z_0)$, where $z_0 \sim \mathcal{N}(0, I_d)$. $U_\alpha(\cdot)$ is parameterized by an expressive neural network such as a Unet (Ronneberger et al., 2015) with parameters $\alpha$. In this way, $p_\alpha(z)$ can be viewed as an implicit generator model.

The molecule generation model $p_\beta(x|z)$ is a conditional autoregressive model, $p_\beta(x|z) = \prod_{t=1}^{T} p_\beta(x^{(t)}|x^{(0)}, ..., x^{(t-1)}, z)$, which is realized by a causal Transformer with parameters $\beta$. The latent vector $z$, serving as the prompt, controls every step of the autoregressive molecule generation. We incorporate these latent prompts $z$ into the $p_\beta(x|z)$ through cross-attention, as shown in Fig. 1.

The real-valued property predictor is a non-linear regression model $p_\gamma(y|z) = \mathcal{N}(s_\gamma(z), \sigma^2)$, where $s_\gamma(z)$ is a small multi-layer perceptron (MLP) predicting $y$ based on the latent prompt $z$. The variance $\sigma^2$ is treated as a hyper-parameter that balances the exploitation-exploration trade-off. For binary-valued constraints, $p_\gamma(y|z) = s_\gamma(z)^y(1 - s_\gamma(z))^{1-y}$. For multi-objective tasks, we can use heuristic-based combinations of multiple objectives to form a special single-objective function.

## 3.2 Maximum Likelihood Learning of LPT

Suppose we observe training examples from the dataset with molecule sequence and property pairs $\mathcal{D} = \{(x_i, y_i), i = 1, ..., n\}$. The log-likelihood function is $L(\theta) = \frac{1}{n} \sum_{i=1}^{n} \log p_\theta(x_i, y_i)$.

Since $z = U_\alpha(z_0)$, we can write the model as

$$p_\theta(x, y) = \int p_\beta(x|z = U_\alpha(z_0)) p_\gamma(y|z = U_\alpha(z_0)) p_0(z_0) dz_0, \tag{4}$$

where $p_0(z_0) \sim \mathcal{N}(0, I_d)$. The learning gradient can be calculated as follows,

$$\nabla_\theta \log p_\theta(x, y) = \mathbb{E}_{p_\theta(z_0|x,y)}[\nabla_\theta \log p_\beta(x|U_\alpha(z_0)) + \nabla_\theta \log p_\gamma(y|U_\alpha(z_0))]. \tag{5}$$

For the prior model, the learning gradient is

$$\delta_\alpha(x, y) = \mathbb{E}_{p_\theta(z_0|x,y)}[\nabla_\alpha \log p_\beta(x|U_\alpha(z_0)) + \nabla_\alpha \log p_\gamma(y|U_\alpha(z_0)))].$$

The learning gradient for the molecule generation model is

$$\delta_\beta(x, y) = \mathbb{E}_{p_\theta(z_0|x,y)}[\nabla_\beta \log p_\beta(x|U_\alpha(z_0))].$$

The learning gradient for the predictor model is

$$\delta_\gamma(x, y) = \mathbb{E}_{p_\theta(z_0|x,y)}[\nabla_\gamma \log p_\gamma(y|U_\alpha(z_0))].$$

Estimating these expectations requires MCMC sampling of the posterior distribution $p_\theta(z_0|x, y)$. We use Langevin dynamics (Neal, 2011). For a target distribution $\pi(z)$, the dynamics iterates as follows,

$$z^{\tau+1} = z^\tau + s\nabla_z \log \pi(z^\tau) + \sqrt{2s}\epsilon^\tau, \tag{6}$$

where $\tau$ indexes the time step, $s$ is step size, and $\epsilon_\tau \sim \mathcal{N}(0, I_d)$ is the Gaussian white noise. Here, $\pi(z)$ is instantiated by the posterior distribution $p_\theta(z_0|x, y)$. With $z = U_\alpha(z_0)$, we have $p(z_0|x, y) \propto p_0(z_0) p_\beta(x|z) p_\gamma(y|z)$. The gradient is

$$\nabla_{z_0} \log p_\theta(z_0|x, y) = \nabla_{z_0} \underbrace{\log p_0(z_0)}_{\text{prior}} + \nabla_{z_0} \underbrace{\sum_{t=1}^{T} \log p_\beta(x^{(t)}|x^{(<t)}, z)}_{\text{autoregressive molecule generation}} + \nabla_{z_0} \underbrace{\log p_\gamma(y|z)}_{\text{property prediction}}.$$

We initialize $z_0^{\tau=0} \sim \mathcal{N}(0, I_d)$, and employ $N$ steps of Langevin dynamics (e.g. $N = 15$) for approximate sampling from the posterior distribution, rendering our learning algorithm as an approximate MLE. See Pang et al. (2020); Nijkamp et al. (2020); Xie et al. (2023) for a theoretical understanding of the learning algorithm based on the finite-step MCMC.

**Pretrain LPT** In practical applications involving multiple molecular generation tasks, each characterized by a different target property $y$, each model $p_\theta(x, y)$ may require separate training. To enhance efficiency, we adopt a pretaining strategy focusing solely on molecule sequences. In this case, we aim to maximize $\sum_{i=1}^{n} \log p_\theta(x_i) = \sum_{i=1}^{n} \log \int p_\theta(x_i, z_i) dz_i$. The learning gradient is $\nabla_\theta \log p_\theta(x) = \mathbb{E}_{p_\theta(z_0|x)}[\nabla_\beta \log p_\theta(x|z = U_\alpha(z_0))]$. After pretraining LPT, we finetune the model with target properties using Eq. (5) for a small number of epochs. This two-stage approach is adaptable for semi-supervised scenarios where property values are limited.

## 3.3 Property Conditioned Generation

After MLE learning from a collected (offline) dataset, LPT is capable of generating molecules with desired properties. Since the latent prompt is designed as the information bottleneck, we can generate the molecule given property value as $p(x|y) = \int p(z_0|y) p(x|z_0) dz_0$. We first infer the latent prompt via posterior sampling using Bayes' rule,

$$z_0 \sim p_\theta(z_0|y) \propto p_0(z_0) p_\gamma(y|z = U_\alpha(z_0)). \tag{7}$$

This posterior sampling is performed using Langevin dynamics similar to the training process. Specifically, we replace the target distribution in Eq. (6) with $p_\theta(z_0|y)$ and run MCMC for a fixed number of steps, i.e.,

$$z_0^{\tau+1} = z_0^\tau + s\nabla_{z_0} \log p(z_0|y) + \sqrt{2s}\epsilon^\tau,$$

$$\text{where} \quad \nabla_{z_0} \log p(z_0|y) = \nabla_{z_0}(\log p(z_0) + \log p(y|z_0)) = -z_0 + \frac{1}{\sigma^2}(y - s_\gamma(z))\nabla_{z_0} s_\gamma(z). \tag{8}$$

This process of sampling $p(z_0|y)$ iteratively refines the latent prompts $z$, increasing their likelihood given the desired property or constraints. This optimization of molecules is achieved in the latent space by gradient-based sampling as a form of test-time computation. Once we generate the latent prompt $z$, the molecule generation model uses this latent prompt to sample the next element $x^{(t)} \sim p_\beta(x^{(t)}|x^{(<t)}, z = U_\alpha(z_0))$ until termination.

MLE learning of the joint model is equivalent to minimizing the $D_{\mathrm{KL}}(p_{\mathrm{data}}(x, y)\|p_\theta(x, y))$. Note that Eq. (7) is reliable when condition $y$ is supported by $p_{\mathrm{data}}(y)$. However, in real-world molecule design, desired property values often lie far from those in the collected offline dataset. For such cases, LPT should be applied in an online learning setting, as discussed in the next section.

### 3.4  Optimization via Online Learning with LPT

When the desired property value $y$ is not supported within the learned distribution $p_\theta(z, x, y)$, we propose an online learning approach to gradually shift the model distribution towards regions supporting desired properties. We assume access to the oracle functions $o(x)$ via software such as RDKit (Landrum et al.), AutoDock-GPU (Santos-Martins et al., 2021).

As discussed in Sec. 3.3, generated molecules are reliable when the desired property value is supported by the learned model. To leverage this, we propose an iterative distribution shifting method, as shown in Alg. 2, as a form of online learning for LPT. In each iteration of distribution shifting, we

(a) Sample molecules from the learned LPT, using incremental extrapolation to generate a synthetic dataset with improved desired properties.
(b) Use hindsight relabeling for the sampled molecules' properties based on oracle functions.
(c) Apply MLE learning of LPT based on this synthetic dataset of molecule-property pairs as described in Sec. 3.2.

This process is repeated until the model converges or budget limits are reached. This online learning method maintains a latent space generative model (LPT) and a synthetic dataset and gradually shifts both towards regions of desired targets.

In step (a), to account for small extrapolation, we use property-conditioned generation as mentioned in Sec. 3.3 by setting the desired property as $y = y^* + \delta_y$, where $y^*$ lies on the boundary of the learned LPT in the previous shifting iteration (e.g. $y^*$ can be the maximum value) and $\delta_y$ is a small increment representing extrapolation. The hyper-parameter $\delta_y$ quantifies the shifting speed. Specifically, since the latent prompts decouple the molecule generation and property prediction, we first sample the latent prompts $z_0 \sim p_\theta(z_0|y = y^* + \delta_y)$ before the molecule generation and then use $x \sim p_\beta(x|z = U_\alpha(z_0))$ to generate the novel molecules. To sample molecules satisfying binary constraints, we use $z_0 \sim p_\theta(z_0|y = 1)$. This latent space sampling scheme relieves the expensive autoregressive sampling in the data space, and allows efficient gradient-based sampling in low-dimensional latent space, such as Langevin dynamics. In step (b), after generating the novel molecules in step (a), we use the oracle function $o(x)$ to relabel them, obtaining a synthetic dataset of molecule-property pairs with ground-truth values. This synthetic dataset serves as a replay buffer, storing generated molecules along the optimization trajectory, and is used for MLE training of the LPT in step (c). The illustration of the online learning of LPT is depicted in Fig. 2.

Compared to population-based methods such as genetic algorithms (Nigam et al., 2020) and particle-swarm algorithms (Winter et al., 2019), our method maintains both a synthetic dataset (which can be considered a small population) and a generative model to fit the dataset, enabling the improvement of generated molecules from the model. The model itself can be seen as an infinite population, as it can generate an unlimited number of new samples.

### 3.5  Efficiency Analysis

In molecule design, computational efficiency and sample efficiency are crucial, with varying priorities depending on the objectives. Computational efficiency measures the ability to learn quickly with limited resources, while sample efficiency assesses the capacity to learn a good model with minimal oracle function interactions. Generative models are applied to molecular design for two purposes: obtaining starting points and optimizing them to meet specific objectives. For objectives accessible via existing software, computational efficiency is prioritized, aiming to train the model and query software as quickly as possible until convergence. For properties that are time-consuming and

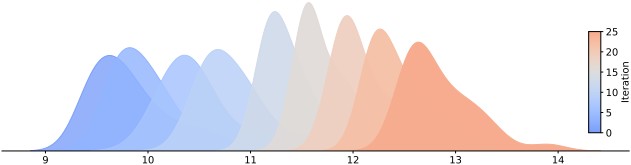

Figure 2: Illustration of online learning LPT. For each shift iteration, we plot the densities of docking scores $E$ using AutoDock-GPU. The increase of the docking scores indicates better binding affinity.

expensive to simulate, such as wet-lab experiments, sample efficiency is more important, minimizing the number of oracle queries for practical usage. In real-world scenarios, a combination of both may be necessary, depending on the specific requirements. We propose the following techniques for LPT to address efficiency issues accordingly.

**Computational efficiency** The computational overhead for online learning of LPT primarily arises from the iterative MCMC sampling procedure. In the posterior sampling stage of $p_\theta(z_0|y)$ and $p_\theta(z_0|x,y)$ in step (a) and (c), rather than using a Gaussian noise-initialized MCMC chain with a fixed number of MCMC steps ($N = 15$) for each learning iteration, we employ the Persistent Markov Chain (PMC) method (Tieleman, 2008; Xie et al., 2016; Han et al., 2017; Xie et al., 2019), which amortizes sampling across shifting iterations. Specifically, for the starting point $z_0^{\tau=0}$ in Eq. (6) of each MCMC chain, its initialization is drawn from a Gaussian distribution at the first learning iteration, then subsequently from the previous iteration's sampling output. With this approach, the number of MCMC updates can be reduced to $N = 2$ steps per sampling stage, achieving an approximate $5\times$ speedup in posterior sampling.

**Sample efficiency** We aim to improve sample efficiency in both steps (a) and (c) in Sec. 3.4. In step (a), property-conditioned generation, we refine the exploration strategy during test-time computation. This approach demonstrates a clear advantage of posterior inference over direct use of VAE or GAN. By guiding the learned LPT towards exploitation during generation, we encourage the latent prompts $z$ to converge on the modes of the posterior distribution in Langevin dynamics. This concept is analogous to the intuition behind classifier guidance in conditional diffusion models (Dhariwal and Nichol, 2021; Ho and Salimans, 2022). In Eq. (8), we can adjust $\sigma^2$ to balance the trade-off between exploitation and exploration: $\nabla_{z_0} \log p(z_0|y) = \nabla_{z_0}(\log p(z_0) + \log p(y|z_0)) = -z_0 + \frac{1}{\sigma^2}(y - s_\gamma(z))\nabla_{z_0} s_\gamma(z)$. When $1/\sigma^2 = 1$, the sampled latent prompts $z$ represent the density of the posterior distribution, resulting in an efficient exploration scheme. As $1/\sigma^2$ increases, the sampled posterior $z$ concentrates around the modes of the posterior distribution, indicating increased confidence and a stronger bias towards exploitation. In step (c), we leverage the synthetic dataset to train the LPT on the most informative samples in terms of property values. By training the LPT on a distribution that assigns higher probability mass to high-value points and lower mass to low-value points, the training objective encourages a larger fraction of the feasible region's volume to model high-value points while simultaneously using other data points to learn useful representations and avoid overfitting. We modify the standard objective $\sum_{i=1}^{n} 1/n \log p_\theta(x_i, y_i)$ by assigning an importance weight $w_i$ to each molecule-property pair $(x_i, y_i)$ in the synthetic dataset, resulting in a biased objective function, $\sum_{i=1}^{n} w_i \log p_\theta(x_i, y_i)$, where $\sum_i w_i = 1$. In our experiments, we set $w_i = 1/N$ if $y_i$ is in the top-$N$ property scores, and $w_i = 0$ otherwise. This approach is inspired by prioritized experience replay (Schaul et al., 2015) in online reinforcement learning and weighted retraining (Tripp et al., 2020) in black-box optimization, both of which prioritize learning from the most informative samples to improve sample efficiency.

## 4 Related Work

**Latent space optimization** Latent space optimization has been widely applied in generative models and high-dimensional data manipulation (Jin et al., 2018; Kusner et al., 2017; Kajino, 2019; Dai et al., 2018). This method involves representing data in a lower-dimensional space while retaining its essential characteristics. Latent space optimization improves the fidelity and quality of generated data and facilitates more efficient and effective exploration of the latent space. This leads to better generalization and robustness in various applications, such as image synthesis (Karras et al., 2019; Razavi et al., 2019; Song et al., 2020; Dhariwal and Nichol, 2021; Rombach et al., 2022), data

compression (Ballé et al., 2016; Mentzer et al., 2018), molecular optimization (Kong et al., 2023; Jain et al., 2023; Zhu et al., 2024), and automatic machine learning (Liu et al., 2018; Zhang et al., 2019). When the search space is significantly simplified in the latent space, familiar Bayesian optimization (BO) tools can be readily applied (Maus et al., 2022; Tripp et al., 2020). Unlike BO-based methods, our LPT is based on the explicit probabilistic form of $p(z|y)$, which allows us to perform optimization as conditional generation using Bayes' rule.

**Generative molecule design** This research follows two main approaches. The first uses latent space generative models to translate discrete molecule graphs into continuous latent vectors, enabling optimization of molecular properties within the latent space (Gómez-Bombarelli et al., 2018; Kusner et al., 2017; Jin et al., 2018; Maziarz et al., 2021; Eckmann et al., 2022; Kong et al., 2023). The second directly employs combinatorial optimization methods, such as reinforcement learning, to fine-tune molecular attributes within the graph data space (You et al., 2018; De Cao and Kipf, 2018; Zhou et al., 2019; Shi et al., 2020; Luo et al., 2021; Du et al., 2022). Alternative data space methodologies, like genetic algorithms (Nigam et al., 2020), particle-swarm strategies (Winter et al., 2019), Monte Carlo tree search (Yang et al., 2020), and scaffolding trees (Fu et al., 2021), have also gained traction.

# 5 Experiments

We demonstrate the effectiveness of our approach across a wide range of optimization tasks. In the context of molecule design, this includes binding affinity maximization, constrained optimization, and multi-objective optimization (see Sec. 5.2). Additionally, we optimize protein sequences for high fluorescence and DNA sequences for enhanced binding affinity (see Sec. 5.3). Finally, we validate the sample efficiency of LPT by performing optimization with a limited number of oracle function queries (see Sec. 5.4). Additional experiments, including robustness to noisy oracles, online learning from scratch, ablation studies and a detailed discussion of related works and baselines, are provided in Apps. A.2 and A.4.

## 5.1 Overview

**Molecule Sequence Design** For molecule design tasks, we use SELFIES representations of the ZINC (Irwin et al., 2012) dataset, which comprises 250K drug-like molecules as our offline dataset $\mathcal{D}$. We utilize RDKit (Landrum et al.) to compute several key metrics, including penalized logP, drug-likeness (QED), and the synthetic accessibility score (SA). Additionally, we use AutoDock-GPU (Santos-Martins et al., 2021) to derive docking scores $E$, which serve as proxies for estimating the binding affinity of compounds to three protein targets: the human estrogen receptor (ESR1), human peroxisomal acetyl-CoA acyltransferase 1 (ACAA1), and Phosphoglycerate dehydrogenase (PHGDH). The binding affinity is expressed as the dissociation constant $K_D$(nM), which is approximated by the formula $K_D \approx e^{-E/c}$, where $E$ is the docking score and $c$ is a constant. A lower $K_D$ value indicates stronger binding affinity. ESR1 is a well-characterized protein with numerous known binders, making it a suitable reference point for evaluating molecules generated by our model. In contrast, ACAA1 has no known binders, providing an opportunity to test the model's capability for *de novo* design (Eckmann et al., 2022). Additionally, we propose to design molecules that bind to PHGDH, an enzyme pivotal in the early stages of L-serine synthesis. Recently, PHGDH has gained attention as a potential therapeutic target in cancer treatment due to to its involvement in various human cancers (Zhao et al., 2020). The crystal structure of PHGDH (PDB: 2G76) is well-established, and there exists a comprehensive case study on the development of PHGDH inhibitors, showcasing a structure-based progression from simpler to more complex molecules targeting its NAD binding site (Spillier and Frédérick, 2021), as illustrated in Fig. 3. Furthermore, we observe that the $K_D$ values estimated from wet lab experiments align closely with the trends predicted by AutoDock-GPU. This congruence makes PHGDH-NAD an excellent case study for testing LPT on single-objective, structure-constrained, and multi-objective optimization tasks using AutoDock-GPU. More details can be found in App. A.5.

**Protein and DNA Sequence Design** We further apply our method to biological sequence design via two tasks in Design-Bench (Trabucco et al., 2022): TF Bind 8 and GFP. To be specific, the TF Bind 8 task focuses on identifying DNA sequences that are 8 bases long, aiming for maximum binding affinity. This task contains a training set of 32,898 samples and includes an exact oracle function. The GFP task involves generating protein sequences of 237 amino acids that exhibit high fluorescence.

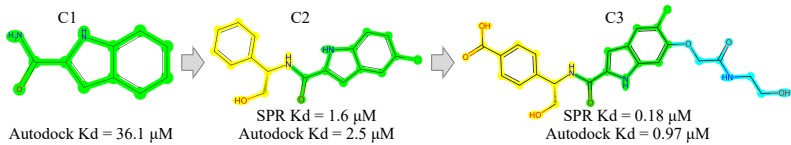

Figure 3: Illustration of the development of PHGDH inhibitors (Spillier and Frédérick, 2021). Surface Plasmon Resonance (SPR) and AutoDock $K_D$ values are reported for each inhibitor. The trends observed between the experimental SPR values and the computational AutoDock values align well, validating the computational approach.

For this task, we use a subset of 5,000 samples as the training set by following the methodology outlined in Trabucco et al. (2022). Due to the unavailability of an exact oracle function for the GFP task, we follow the same oracle function preparation as in Design-Bench, and train a Transformer regression model on the full dataset with a total of 56,086 samples as the oracle function. We evaluate the design performance and diversity of the generated samples.

**Training Setup** The prior model, $p_\alpha(z)$, of LPT is a one-dimensional UNet (Ronneberger et al., 2015) where $z$ contains 4 tokens, each of size 256. The sequence generation model, $p_\beta(x|z)$, is implemented as a 3-layer causal Transformer, while a 3-layer MLP serves as the predictor model, $p_\gamma(y|z)$. As described in Sec. 3.2, we pre-train LPT on molecules for 30 epochs and then fine-tune it with target properties for an additional 10 epochs, following the procedure outlined in Alg. 1 in App. A.3. We perform up to 25 iterations of online learning, generating 2,500 samples per iteration, which totals a maximum of 62.5K oracle function queries. We use the AdamW optimizer (Loshchilov and Hutter, 2019; Kingma and Ba, 2014) with a weight decay of 0.1. Training was conducted on an NVIDIA A6000 GPU, requiring 20 hours for pre-training, 10 hours for fine-tuning, and 12 hours for online learning. Additional details can be found in App. A.3.

## 5.2 Binding Affinity Maximization

**Single-Objective Optimization** For the single-objective binding affinity optimization task, we aim to design ligands with optimal binding affinities to ESR1, ACAA1, and PHGDH as *de novo* design, without any constraints. LPT does not use any prior knowledge of existing binders, and exclusively uses the crystal structures of the aforementioned proteins. The predictor model $p_\gamma(y|z)$ for this task is a regression model that estimates docking scores. We compare our model with several baseline methods, which are introduced in App. A.4. As shown in Tabs. 1 and 2, our model significantly surpasses other methods across all three binding affinity maximization tasks in terms of $K_D$, often achieving substantial improvement. Lower $K_D$ values indicate better performance. Furthermore, in Tab. 2, we report the average performance of the top 50 and top 100 molecules to demonstrate that our model can effectively generate a diverse pool of candidate molecules with the desired properties. Visualizations of generated molecules are provided in App. A.6.1.

Table 1: Single-objective binding affinity optimization results for ESR1 and ACAA1. Top 3 performance in terms of $K_D(\text{nM})$ achieved by each model are reported. The best scores are in bold.

| METHOD | ESR1 $K_D$ ($\downarrow$) | | | ACAA1 $K_D$ ($\downarrow$) | | |
|---|---|---|---|---|---|---|
| | 1ST | 2RD | 3RD | 1ST | 2RD | 3RD |
| GCPN | 6.4 | 6.6 | 8.5 | 75 | 83 | 84 |
| MOLDQN | 373 | 588 | 1062 | 240 | 337 | 608 |
| MARS | 25 | 47 | 51 | 370 | 520 | 590 |
| GRAPHDF | 17 | 64 | 69 | 163 | 203 | 236 |
| LIMO | 0.72 | 0.89 | 1.4 | 37 | 37 | 41 |
| SGDS | 0.03 | 0.03 | 0.04 | 0.11 | 0.11 | 0.12 |
| **LPT** | **0.004** | **0.005** | **0.009** | **0.037** | **0.041** | **0.045** |

Table 2: Single-objective binding affinity maximization results for PHGDH, reporting the 1st, 2nd, and 3rd performance, along with average performance of top 50 and top 100 molecules for each model. Performance is measured by $K_D(10^{-2}\text{nM})$. The highest scores are in bold.

| | PHGDH $K_D$ ($\downarrow$) | | |
|---|---|---|---|
| | LIMO | SGDS | LPT |
| 1ST | 13.87 | 3.65 | **0.07** |
| 2ND | 18.79 | 6.59 | **0.16** |
| 3RD | 21.15 | 7.16 | **0.19** |
| TOP-50 | 81.16 ± 34.41 | 14.7 ± 5.43 | **0.95 ± 0.46** |
| TOP-100 | 125.06 ± 52.59 | 24.5 ± 14.0 | **1.66 ± 0.82** |

**Structure-constrained Optimization** The structure-constrained optimization task mimics lead optimization in drug discovery, aiming to decorate a fixed core substructure to optimize activity and pharmacological properties. Our model's factorization, $p(z, x, y) = p(z)p(x|z)p(y|z)$, enables the decoupling of molecule generation and property prediction, simplifying conditional generation.

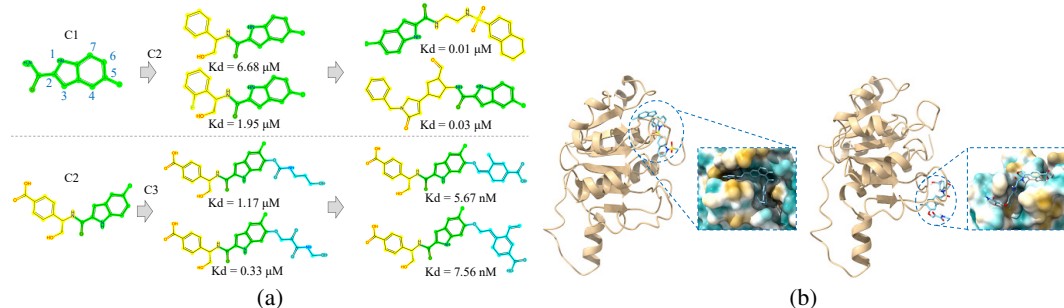

(a)                          (b)

Figure 4: (a) Structure-constrained Optimization. Conditionally generated compounds C2 and C3 closely resemble the human-designed compounds C2 and C3 shown in Fig. 3. Additionally, the right column also presents further optimized compounds that achieve improved $K_D$ scores. (b) Illustration of generated molecules binding to PHGDH with docking poses generated by AutoDock-GPU. The left panel visualizes the molecule generated through multi-objective optimization, while the right panel displays the molecule generated via structure-constrained optimization.

Given a substructure $\hat{x} = (x^{(1)}, \ldots, x^{(k)})$, we aim to sample from $p_\theta(x, y|\hat{x})$. This is accomplished by sampling $z \sim p(z|y)$ and then $x \sim p(x|\hat{x}, z)$, which only requires rearranging $\hat{x}$'s sequence to start from the desired atom. In Fig. 3, Compound 2 (C2) is designed by humans as an extension of Compound 1 (C1), and Compound 3 (C3) is similarly an extension of C2. In Fig. 4, we show that: (1) given C1 or C2, our model is able to design compounds similar to the human-designed C2 or C3, and (2) our model can identify molecules that outperform those designed by humans. Meanwhile, we confirm that C2, C3, and LPT-generated molecules are *novel* compared to the ZINC training set, with an average Tanimoto similarity less than $0.5$. More generated molecules are shown in App. A.6.2.

Our model explores high-affinity PHGDH inhibitors by adding functional groups to an indole backbone, a common scaffold for such inhibitors (Spillier and Frédérick, 2021). The model identifies aromatic or heteroaromatic groups, such as benzene or pyridine, at the second position as frequently occurring and exhibiting higher binding scores compared to the indole backbone itself. This finding aligns with reported data: a published molecule with a benzothiophene backbone similar to C1 exhibits a binding affinity of 470 $\mu$M, while C2, which includes an aromatic group at the second position, shows a significantly improved binding affinity of 1.6 $\mu$M (Spillier and Frédérick, 2021; Fuller et al., 2016). Furthermore, the model introduces a second functional group at the sixth position of the indole in C2, generating inhibitors closely resembling the structure of C3. The observed trend of increasing binding affinities (1.6 $\mu$M for C2 and 0.18 $\mu$M for C3) aligns with the literature values, providing validation for our proposed method in identifying potential high-affinity inhibitors.

**Multi-Objective Optimization** For multi-objective optimization tasks, we aim to simultaneously maximize binding affinity and QED, while minimizing SA. These objectives are balanced as a weighted combination, with constraints of QED $> 0.4$ and SA $< 5.5$. We evaluate our method on three protein targets: ESR1, ACAA1, and PHGDH, comparing the results against two baseline methods, LIMO (Eckmann et al., 2022) and SGDS (Kong et al., 2023). As shown in Tab. 3, our method, LPT, achieves QED and SA scores comparable to those of SGDS while significantly improving binding affinity across all three protein targets, demonstrating its superior modeling capability. Examples of the generated molecules are provided in App. A.6.1.

Table 3: Multi-objective optimization Results. Top 2 performance, measured by $K_D$(nM), QED and SA, are reported for each method. Baseline methods include LIMO (Eckmann et al., 2022) and SGDS (Kong et al., 2023). Best results are marked in bold, and the second best results are underlined.

| LIGAND | ESR1 | | | ACAA1 | | | PHGDH | | |
|---|---|---|---|---|---|---|---|---|---|
| | $K_D \downarrow$ | QED $\uparrow$ | SA $\downarrow$ | $K_D \downarrow$ | QED $\uparrow$ | SA $\downarrow$ | $K_D \downarrow$ | QED $\uparrow$ | SA $\downarrow$ |
| LIMO 1ST | 4.6 | 0.43 | 4.8 | 28 | 0.57 | 5.5 | 29.15 | 0.33 | 4.73 |
| LIMO 2ND | 2.8 | **0.64** | 4.9 | 31 | 0.44 | 4.9 | 42.98 | 0.20 | 5.32 |
| SGDS 1ST | 0.36 | 0.44 | 3.99 | 4.55 | 0.56 | **4.07** | 4.47 | **0.54** | 3.37 |
| SGDS 2ND | 1.28 | 0.44 | 3.86 | 5.67 | 0.60 | 4.58 | 5.39 | 0.42 | 4.02 |
| **LPT 1ST** | **0.04** | 0.58 | 3.46 | **0.18** | 0.50 | 4.85 | **0.02** | 0.50 | **3.11** |
| **LPT 2ND** | 0.05 | 0.46 | **3.24** | 0.21 | **0.61** | 4.18 | 0.03 | 0.43 | 3.22 |

## 5.3 Biological Sequence Design

Our model excels in biological sequence design, a single-objective optimization application, as demonstrated by two benchmarks in Design-Bench (Trabucco et al., 2022): TF Bind 8 and GFP. Tab. 4 shows that LPT significantly outperforms other methods in these tasks. In the TF Bind 8 task, our approach surpasses the strong competitor GFlowNet-AL (Jain et al., 2022), while maintaining comparable diversity. For the GFP task, we achieve superior performance with reasonable diversity.

Table 4: Results of biological sequence design on TF Bind 8 and GFP benchmarks. Performance and diversity are evaluated on 128 samples. Results of other baselines are obtained from Jain et al. (2022). Bold highlighting indicates top scores.

| METHOD | TF BIND 8 | | GFP | |
|---|---|---|---|---|
| | PERFORMANCE | DIVERSITY | PERFORMANCE | DIVERSITY |
| DYNAPPO | $0.58 \pm 0.02$ | $5.18 \pm 0.04$ | $0.05 \pm 0.008$ | $12.54 \pm 1.34$ |
| COMs | $0.74 \pm 0.04$ | $4.36 \pm 0.24$ | $0.831 \pm 0.003$ | $8.57 \pm 1.21$ |
| BO-qEI | $0.44 \pm 0.05$ | $4.78 \pm 0.17$ | $0.045 \pm 0.003$ | $12.87 \pm 1.09$ |
| CbAS | $0.45 \pm 0.14$ | $5.35 \pm 0.16$ | $0.817 \pm 0.012$ | $8.53 \pm 0.65$ |
| MINs | $0.40 \pm 0.14$ | $\mathbf{5.57 \pm 0.15}$ | $0.761 \pm 0.007$ | $8.31 \pm 0.02$ |
| CMA-ES | $0.47 \pm 0.12$ | $4.89 \pm 0.01$ | $0.063 \pm 0.003$ | $10.52 \pm 4.24$ |
| AMORTIZEDBO | $0.62 \pm 0.01$ | $4.97 \pm 0.06$ | $0.051 \pm 0.001$ | $16.14 \pm 2.14$ |
| GFLOWNET-AL | $0.84 \pm 0.05$ | $4.53 \pm 0.46$ | $0.05 \pm 0.010$ | $\mathbf{21.57 \pm 3.73}$ |
| **LPT** | $\mathbf{0.954 \pm 0.002}$ | $4.58 \pm 0.06$ | $\mathbf{0.857 \pm 0.003}$ | $9.45 \pm 0.23$ |

## 5.4 Sample Efficiency

We validate LPT's sample efficiency on the Practical Molecular Optimization (PMO) benchmark (Gao et al., 2022), where multi-property objectives (MPO) are optimized within a limited oracle budget of 10K queries. Tab. 5 shows that our method, LPT, surpasses previous approaches, such as MARS (Xie et al., 2021), GFlowNet (Jain et al., 2022) and SMILES/SELFIES-VAE (Gómez-Bombarelli et al., 2018; Maus et al., 2022) with Bayesian Optimization (BO). Also, LPT achieves comparable performance to LSTM HC (Brown et al., 2019), the best generative molecule design method in PMO, and demonstrates performance on par with GP-BO (Tripp et al., 2021), the best BO-based method in PMO, under the limited budge of oracle function queries. We acknowledge that there remains a performance gap between generative model-based optimization and methods like genetic algorithms when working with a small budget. This is primarily due to the data-intensive requirements of training generative models. To ensure a fair assessment, our comparison focuses on representative generative molecule design methods within PMO. It's worth noting that generative models offer distinct advantages when maintaining relatively large budgets, as the learned model itself can be viewed as infinite populations for further exploration.

Table 5: Comparison of sample efficiency on the PMO benchmark. The mean and standard deviation of AUC Top-10 from 5 independent runs are reported. Best results are marked in bold.

| METHOD | AMLODIPINE | FEXOFENADINE | OSIMERTINIB | PERINDOPRIL | RANOLAZINE | ZALEPLON | SUM |
|---|---|---|---|---|---|---|---|
| GFLOWNET | $0.444 \pm 0.004$ | $0.693 \pm 0.006$ | $0.784 \pm 0.001$ | $0.430 \pm 0.010$ | $0.652 \pm 0.002$ | $0.035 \pm 0.030$ | 3.038 |
| MARS | $0.504 \pm 0.016$ | $0.711 \pm 0.006$ | $0.777 \pm 0.006$ | $0.462 \pm 0.006$ | $\mathbf{0.740 \pm 0.010}$ | $0.187 \pm 0.046$ | 3.381 |
| LSTM HC | $0.593 \pm 0.016$ | $\mathbf{0.725 \pm 0.003}$ | $\mathbf{0.796 \pm 0.002}$ | $0.489 \pm 0.007$ | $0.714 \pm 0.008$ | $0.206 \pm 0.006$ | 3.523 |
| SMILES-VAE BO | $0.533 \pm 0.009$ | $0.671 \pm 0.003$ | $0.771 \pm 0.002$ | $0.442 \pm 0.004$ | $0.457 \pm 0.012$ | $0.039 \pm 0.012$ | 2.913 |
| SELFIES-VAE BO | $0.516 \pm 0.005$ | $0.670 \pm 0.004$ | $0.765 \pm 0.002$ | $0.429 \pm 0.003$ | $0.452 \pm 0.025$ | $0.206 \pm 0.015$ | 3.038 |
| GP-BO | $0.583 \pm 0.044$ | $0.722 \pm 0.005$ | $0.787 \pm 0.006$ | $0.493 \pm 0.011$ | $0.735 \pm 0.013$ | $0.221 \pm 0.072$ | 3.541 |
| **LPT** | $\mathbf{0.608 \pm 0.005}$ | $0.714 \pm 0.003$ | $0.784 \pm 0.011$ | $\mathbf{0.511 \pm 0.002}$ | $0.682 \pm 0.007$ | $\mathbf{0.245 \pm 0.003}$ | 3.544 |

## 6 Limitation and Conclusion

In this work, we presented LPT, a novel generative model for molecule design that achieves strong performance through its offline and online learning algorithms. Contemporary work has extended the similar model to offline reinforcement learning (Kong et al., 2024). While the model demonstrates significant potential, there are opportunities to better understand how LPT handles the inherent trade-offs in multi-objective optimization scenarios, particularly in characterizing the Pareto front nature of optimal solutions. Future work could also explore alternative architectures to extend LPT's applicability beyond sequence-based optimization problems in science and engineering.

## Acknowledgements

The authors would like to thank the anonymous reviewers for providing constructive comments and suggestions to improve the work. The work was partially supported by NSF DMS-2015577, NSF DMS-2415226, a gift fund from Amazon, and XSEDE grant CIS210052.

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

# A   Appendix

## A.1   Model Learning

We provide the derivation of Eq. (5) in Sec. 3.2.

$$
\begin{aligned}
\nabla_\theta \log p_\theta(x,y) &= \frac{\nabla_\theta p_\theta(x,y)}{p_\theta(x,y)} \\
&= \frac{1}{p_\theta(x,y)} \int \nabla_\theta p_\theta(x,y,z=U_\alpha(z_0)) dz_0 \\
&= \int \frac{p_\theta(x,y,z=U_\alpha(z_0))}{p_\theta(x,y)} \nabla_\theta \log p_\theta(x,y,z=U_\alpha(z_0)) dz_0 \\
&= \int p_\theta(z_0|x,y) \nabla_\theta \log p_\theta(x,y,z=U_\alpha(z_0)) dz_0 \\
&= \mathbb{E}_{p_\theta(z_0|x,y)} \left[ \nabla_\theta \log p_\theta(x,y,z=U_\alpha(z_0)) \right] \\
&= \mathbb{E}_{p_\theta(z_0|x,y)} \left[ \nabla_\theta \log p_\beta(x|U_\alpha(z_0)) + \nabla_\theta \log p_\gamma(y|U_\alpha(z_0)) + \nabla_\theta \log p_0(z_0) \right] \\
&= \mathbb{E}_{p_\theta(z_0|x,y)} \left[ \nabla_\theta \log p_\beta(x|U_\alpha(z_0)) + \nabla_\theta \log p_\gamma(y|U_\alpha(z_0)) \right].
\end{aligned}
$$

## A.2   Additional Experiments

### A.2.1   Model Sanity Check: Penalized logP and QED Maximization

The experiments focus on optimizing the Penalized logP (P-logP) and QED properties, both of which can be calculated using RDKit. Since P-logP scores are positively correlated with the length of a molecule, we maximize P-logP while limiting molecule length to the maximum length of molecules in ZINC using the SELFIES (Krenn et al., 2020; Cheng et al., 2023) representation, following Eckmann et al. (2022). We compare our model with several baseline methods, including JT-VAE (Jin et al., 2018), MolDQN (Zhou et al., 2019), LIMO (Eckmann et al., 2022), GCPN (You et al., 2018), GraphDF (You et al., 2018), MARS (Xie et al., 2021), SGDS (Kong et al., 2023). Tab. 6 presents the results, demonstrating that LPT outperforms other methods and achieves the highest QED score among methods that limits molecule length.

Table 6: Results of P-logP and QED maximization. The top 3 highest scores achieved by each model are reported. "Length Limit" indicates the application of a maximum molecule length limit.

| METHOD | LENGTH LIMIT | PENALIZED LOGP ($\uparrow$) | | | QED ($\uparrow$) | | |
|---|---|---|---|---|---|---|---|
| | | 1ST | 2RD | 3RD | 1ST | 2RD | 3RD |
| JT-VAE | ✗ | 5.30 | 4.93 | 4.49 | 0.925 | 0.911 | 0.910 |
| GCPN | ✓ | 7.98 | 7.85 | 7.80 | **0.948** | 0.947 | 0.946 |
| MOLDQN | ✓ | 11.8 | 11.8 | 11.8 | **0.948** | 0.943 | 0.943 |
| MARS | ✗ | 45.0 | 44.3 | 43.8 | **0.948** | **0.948** | **0.948** |
| GRAPHDF | ✗ | 13.7 | 13.2 | 13.2 | **0.948** | **0.948** | **0.948** |
| LIMO | ✓ | 10.5 | 9.69 | 9.60 | 0.947 | 0.946 | 0.945 |
| SGDS | ✓ | 26.4 | 25.8 | 25.5 | **0.948** | **0.948** | **0.948** |
| **LPT** | ✓ | **38.95** | **38.29** | **38.25** | **0.948** | **0.948** | **0.948** |

### A.2.2   Robustness to Noisy Oracles

To evaluate the robustness of our model in single-objective QED optimization tasks under the oracle query budget of 25K, we conduct experiments with varying levels of oracle noise. We define noised oracles as $y_{\text{noise}} = y_{\text{true}} + e$, where $e \sim \mathcal{N}(0, \sigma^2)$ and $\sigma$ varies as a percentage of the property range. The minimal degradation in performance seen in Tab. 7 demonstrates that our model is resilient to the noised oracles.

### A.2.3   Online Learning from Scratch

We investigate LPT's performance without offline pre-training data, relying solely on online learning with a budget limit of 300K (comparable to the size of the ZINC dataset plus our previous online

Table 7: Performance across different Oracle noise levels

| ORACLE NOISE | 1ST | 2ND | 3RD | TOP-50 |
|---|---|---|---|---|
| NONE | 0.948 | 0.947 | 0.947 | $0.940\pm0.003$ |
| 1% | 0.947 | 0.947 | 0.946 | $0.939\pm0.004$ |
| 5% | 0.946 | 0.946 | 0.944 | $0.936\pm0.005$ |
| 10% | 0.945 | 0.945 | 0.942 | $0.932\pm0.006$ |

learning budget). Results in Tab. 8 show that LPT can effectively discover high-binding molecules for ESR1 and ACAA1, even without pre-training. However, performance on PHGDH remained suboptimal compared to the version with pre-training, indicating that this target may require additional oracle queries due to its inherent complexity. These findings highlight the potential of pure online learning approaches for future exploration.

Table 8: Results of online learning from scratch. Top 3 scores of $K_D(\text{nM})$ are reported.

| | 1ST | 2ND | 3RD |
|---|---|---|---|
| ESR1 | 0.012 | 0.019 | 0.021 |
| ACAA1 | 0.047 | 0.062 | 0.128 |
| PHGDH | 0.158 | 0.339 | 0.342 |

#### A.2.4 Ablation Studies

We conduct ablation studies to investigate the contributions of key components in LPT using a challenging PHGDH single-objective optimization task. The variations in our experiments included:

1. Using samples from $z \sim p_\alpha(z)$ instead of $p_\theta(z|y)$ to generate the proposals $\mathcal{P}^t$.

2. Removing weighted retraining and applying the standard objective $\sum_{i=1}^{n} {}^1\!/n \log p_\theta(x_i, y_i)$.

3. Setting the total number of shifting iterations as 1.

4. Replacing the Unet prior with a Gaussian $\mathcal{N}(0, I_d)$.

As shown in Tab. 9, each component was essential, and removing or altering any of them might lead to performance degradation, underscoring their importance in achieving the model's high efficacy.

Table 9: Ablation of Key Components. We report the mean and standard deviation of $K_D(\text{nM})$ over the top 100 unique molecules generated in the last shifting iteration.

| METHOD | TOP 100 |
|---|---|
| LPT | $0.08 \pm 0.04$ |
| SAMPLING FROM $z \sim p_\alpha(z)$ | $403.18 \pm 246.54$ |
| STANDARD OBJECTIVE $\sum_{i=1}^{n} {}^1\!/n \log p_\theta(x_i, y_i)$ | $365.35 \pm 144.94$ |
| NUMBER OF SHIFTING ITERATION AS 1 | $41.55 \pm 19.02$ |
| WITHOUT LEARNABLE PRIOR | $417.22 \pm 249.57$ |

In addition, we investigate the effect of the exploitation scheme ${}^1\!/\sigma^2$ in Eq. (8). Experiments are conducted on the practical molecular optimization (PMO) benchmark, focusing on optimizing the multi-property objective (MPO) for amlodipine under a 10K-query oracle budget. As shown in Tab. 10, increasing ${}^1\!/\sigma^2$ enhances LPT's performance, indicating a stronger bias toward exploiting the sampled posterior $z$ and leading to improved optimization efficiency.

Furthermore, we study the effect of oracle budget size on performance, conducting experiments on the single-objective ESR1 binding affinity optimization task. As shown in Tab. 11, LPT demonstrates robust performance even under a limited oracle query budget of 10k, outperforming most existing methods. As the budget increases, LPT continues to exhibit significant performance enhancements, further highlighting its efficiency and scalability.

Table 10: Effects of the exploration schemes. We report the AUC Top-10 for the multi-property objective (MPO) on amlodipine in the PMO benchmark.

| EXPLORATION SCHEME $1/\sigma^2$ | AUC TOP-10 |
|---|---|
| 1 | 0.529 |
| 5 | 0.536 |
| 20 | 0.554 |
| 40 | 0.583 |
| 80 | 0.608 |

Table 11: Single-objective ESR1 binding affinity $K_D(\text{nM})$ optimization with different budgets.

| BUDGETS | 1ST | 2ND | 3RD |
|---|---|---|---|
| 10K | 4.5 | 4.8 | 5.5 |
| 30K | 1.4 | 1.7 | 2.1 |
| 50K | 0.08 | 0.10 | 0.11 |
| 62.5K | 0.04 | 0.05 | 0.06 |

## A.3 Model Architecture and Training Details

As illustrated in Fig. 1, the prior model in LPT is parameterized by a 1D Unet, with $z$ sequence length of 4, where each of them is size of 256. The molecule generation model, $p_\beta(x|z)$, employs a 3-layer causal Transformer with an embedding size of 256 and maximum input token length of 73. The predictor model is a 3-layer MLP, which takes $z$ as input and outputs predicted property values or classification results. The total number of parameters for LPT is 4.33M.

LPT is trained in a two-step process. Initially, it undergoes pre-training solely on molecules for 30 epochs, using cross-entropy loss with a learning rate that varies between $7.5e-4$ and $7.5e-5$, following a cosine scheduling approach. Subsequently, LPT is fine-tuned for 10 epochs on both molecules and their properties, as outlined in Alg. 1, with the learning rate adjusted between $3e-4$ and $7.5e-5$. For multi-objective optimization, i.e., simultaneously maximizing binding affinity, QED, and minimizing SA, the predictors for binding affinity and QED/SA are selected as a regression model and a classifier, which are supervised by mean squared error (MSE) and binary cross-entropy (BCE) loss functions, respectively.

---

**Algorithm 1** MLE learning of Latent Prompt Transformer (LPT)

---

**Input:** Number of learning iterations $T$, initial parameters $\theta_0 = (\alpha_0, \beta_0, \gamma_0)$, observed samples $\mathcal{D} = \{x_i, y_i\}_{i=1}^n$, posterior sampling step size $s$, the number of MCMC steps $N$, and the learning rate $\eta_0, \eta_1, \eta_2$.
**Output:** $\theta_T$
**for** $t = 1$ **to** $T$ **do**
    1.**Posterior sampling**: For each $(x_i, y_i)$, sample $z_0 \sim p_{\theta_t}(z_0|x_i, y_i)$ using Eq. (6), where the target distribution $\pi$ is $p_{\theta_t}(z_0|x_i, y_i)$ with $N$ steps and step size $s$.
    2.**Learn prior model** $p_\alpha(z)$, **generation model** $p_\beta(x|z)$ and **predictor model** $p_\gamma(y|z)$:
    $\alpha_{t+1} = \alpha_t + \eta_0 \frac{1}{n} \sum_i \delta_\alpha(x_i, y_i)$;
    $\beta_{t+1} = \beta_t + \eta_1 \frac{1}{n} \sum_i \delta_\beta(x_i, y_i)$;
    $\gamma_{t+1} = \gamma_t + \eta_2 \frac{1}{n} \sum_i \delta_\gamma(x_i, y_i)$ as in Sec. 3.2.
**end for**

---

For LPT's online learning, as detailed in Alg. 2, we establish a maximum number of shifting iterations to be 25, generating 2,500 new samples in each iteration. This results in a total of up to 62.5k oracle function queries. Throughout the training processes, we utilize the AdamW optimizer (Loshchilov and Hutter, 2019) with a weight decay of 0.1. The pre-training, fine-tuning, and online learning phases of LPT require approximately 20, 10, and 12 hours, respectively, on a single NVIDIA A6000 GPU.

**Algorithm 2** Online learning of LPT

---

**Input:** Number of proposals $m$, fixed increment $\delta_y$, number of PMC steps $N$, initial parameters $\theta_t = (\alpha_t, \beta_t, \gamma_t)$, initial dataset $\mathcal{D}^0 = \{x_i^0, y_i^0\}_{i=1}^n$, oracle function $o(x)$, maximum number of shifting iterations $T$.
**Output:** $\theta^T, \mathcal{D}^T$
**while** $t < T$ **do**

    step (a): Sample molecules and properties from the learned model $\{z_i^t, x_i^t, y_i^t\}_{i=1}^m$, where $z^t \sim p_\theta(z|y = y^{t-1} + \delta_y)$; $x^t \sim p_\beta(x|z = z^t)$.

    step (b): Relabel the proposal property values by oracle function query: $y_i^t \leftarrow o(x_i^t)$, and update the dataset by $\mathcal{D}^t = \{z_i^t, x_i^t, y_i^t\}_{i=1}^m \cup \mathcal{D}^{t-1}$.

    step (c): Update LPT using maximum likelihood on synthetic dataset $\mathcal{D}^t$ by following Alg. 1

**end while**

---

## A.4 Baselines

Our model is based on Kong et al. (2023). The differences are as follows. (1) While Kong et al. (2023) used an LSTM model for molecule generation, we adopt a more expressive causal Transformer model for generation, with the latent vector serving as latent prompt. (2) While Kong et al. (2023) used a latent space energy-based model for the prior distribution of the latent vector, we assume that the latent $z$ is generated by a UNet transformation of a Gaussian white noise vector. This approach allows us to eliminate the need for Langevin dynamics in prior sampling during training, thus simplifying the learning algorithm. (3) Our experimental results are significantly stronger, surpassing those of Kong et al. (2023) and achieving new state-of-the-art performance.

For molecule generation, JT-VAE (Jin et al., 2018) utilizes a variational autoencoder (VAE). GCPN (You et al., 2018) and GraphDF (Luo et al., 2021) employ the deep graph model to discover novel molecules. A reinforcement learning framework that fuses chemical domain knowledge with double Q-learning and randomized value functions for molecule optimization was presented by MolDQN (Zhou et al., 2019). MARS (Xie et al., 2021) develops a Markov molecular Sampling framework targeting multi-objective drug discovery, while LIMO (Eckmann et al., 2022) uses a VAE-generated latent space along with neural networks for property prediction, enabling efficient gradient-based optimization of molecular properties.

In contrast to existing latent space generative models (Gómez-Bombarelli et al., 2018; Kusner et al., 2017; Jin et al., 2018; Eckmann et al., 2022), our approach incorporates a learnable prior model, enabling our model to effectively catch up with the evolving dataset in the optimization process.

The landscape of biological sequence design is shaped by a diverse range of computational strategies. DyNAPPO (Angermueller et al., 2019) harnesses active learning with reinforcement learning to facilitate iterative sequence generation. GFlowNet-AL (Jain et al., 2022) capitalizes on GFlowNets for generative active learning within sequence contexts. Model-based optimization techniques like COMs (Trabucco et al., 2021) and AmortizedBO (Swersky et al., 2020) integrate Bayesian Optimization with reinforcement learning, enhancing search efficiency. BO-QEI (Wilson et al., 2017), a variant of Bayesian optimization, refines the search process using quantile Expected Improvement. Deep generative models, e.g., CBAs (Fannjiang and Listgarten, 2020) and MINs (Brookes et al., 2019), leverage deep learning for complex pattern discovery. Meanwhile, CMA-ES (Hansen, 2006) excels in high-dimensional optimization, adapting search strategies over generations.

## A.5 Background of PHGDH and its NAD binding site

Phosphoglycerate dehydrogenase (PHGDH) is an enzyme that plays a crucial role in the early stages of L-serine synthesis. Recently, PHGDH has been identified as an attractive therapeutic target in cancer therapy due to its involvement in various human cancers, including breast cancer, melanoma, lung cancer, pancreatic cancer, and kidney cancer. Several studies have been devoted to the exploration of small molecule inhibitors targeting PHGDH including CBR-5884 (IC50 = 33±12 ($\mu$M)), NCT-503 (IC50 = 2.5±0.6 ($\mu$M)), AZ PHGDH inhibitor (IC50 = 180 (nM)), RAZE PHGDH inhibitor (IC50 = 0.01 $\sim$ 1.5 ($\mu$M)), PKUMDL-WQ-2201 (IC50 = 35.7 ($\mu$M)), BI-4924 (IC50 = 2 (nM)), and others. Additionally, the crystal structure of PHGDH (PDB: 2G76) has been elucidated.

This makes PHGDH an excellent protein model for conducting docking calculations, aiding in the precise localization of binding sites in order to optimize our algorithms.

PHGDH utilizes nicotinamide adenine dinucleotide (oxidized form, NAD+; reduced form, NADH) as a co-factor for enzymatic activity, producing NADH during the synthesis of 3-phosphohydroxypyruvate (3PHP) from 3-phosphoglycerate (3PG) (Samanta and Semenza, 2016).

The co-crystallization of PHGDH with NAD has been documented (PDB: 5N6C). The NAD+ pocket is surrounded by hydrophobic residues of P176, Y174, L151, L193, L216, T213, T207 and L210 (Mullarky et al., 2019). Specifically, nicotinamide moiety exhibited interactions with the protein backbone (specifically A285 and C233) as well as the side chain of D259. Additionally, the hydroxyl groups of the sugar moieties were also involved in hydrogen bonds with both the protein backbone (T206) and the side chain of D174. The phosphate linker demonstrated interactions with the main chain of R154 and I155, along with the side chain of R154 (Fig. 5) (Unterlass et al., 2017).

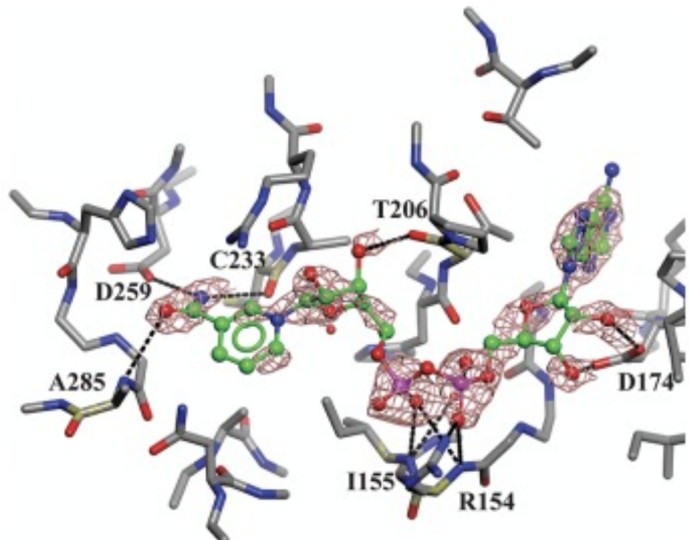

Figure 5: PHGDH with NAD binding site.

## A.6 Visualization of Generated Molecules

We present visualizations of a subset of generated molecules from both single-objective and multi-objective optimization for ESR1, ACAA1, and PHGDH. Additionally, several conditionally generated compounds, C2 and C3, which have similar characteristics to human-designed ones, are also presented in Fig. 12 and Fig. 13.

### A.6.1 Multi-Objective and Single-Objective Optimization

Figs. 6 to 8 visualize some molecules produced during the multi-objective optimization for ESR1, ACAA1, and PHGDH, respectively. Meanwhile, Figs. 9 to 11 depict some examples of molecules produced during the single-objective optimization for ESR1, ACAA1, and PHGDH, respectively.

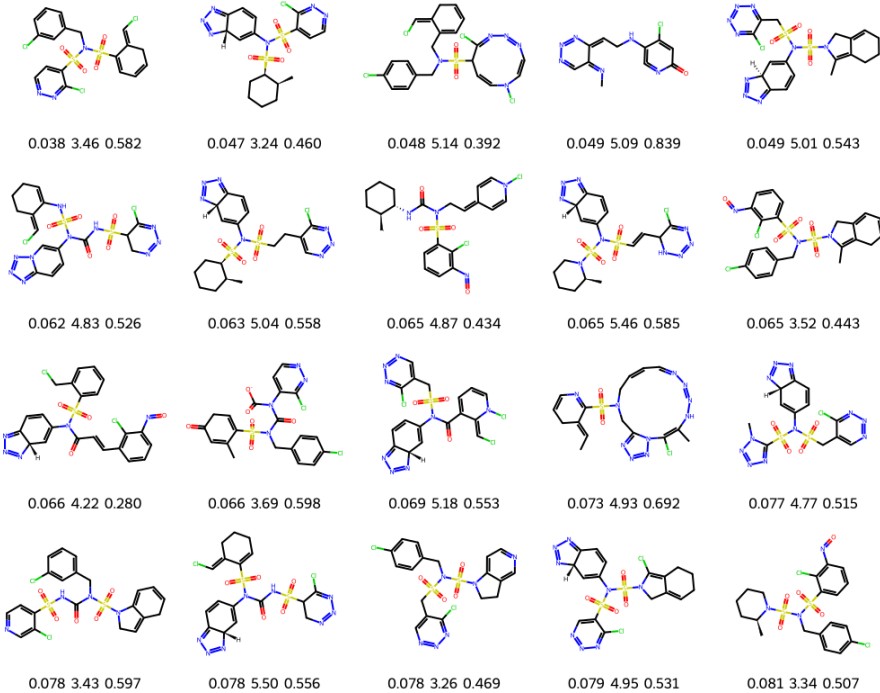

Figure 6: Molecules produced during the multi-objective optimization for ESR1. The legends denote $K_D(\mathrm{nM}) \downarrow$, SA↓ and QED↑.

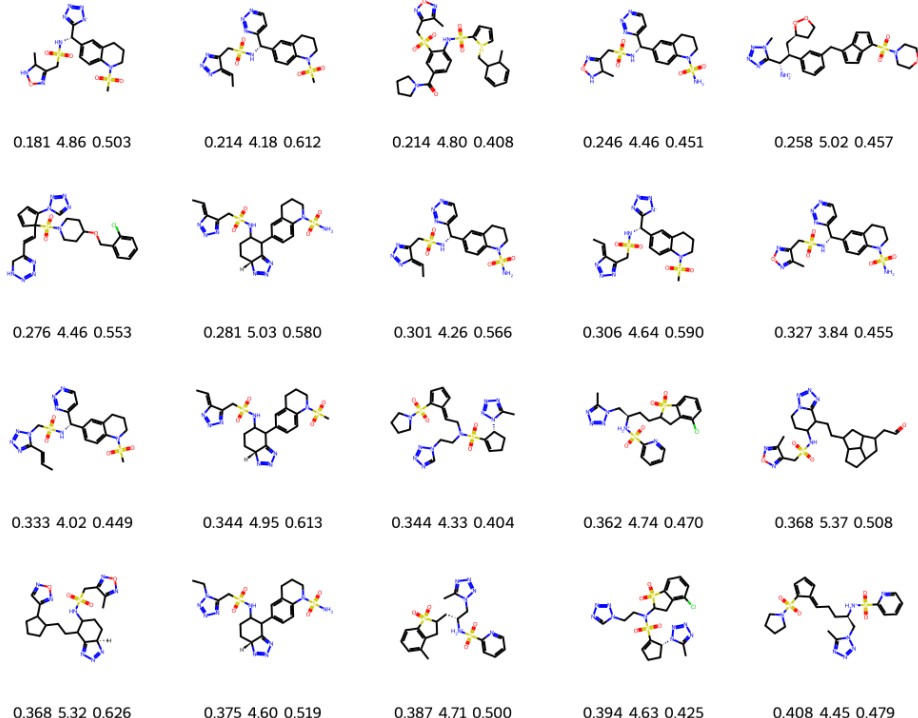

Figure 7: Molecules produced during the multi-objective optimization for ACAA1. The legends denote $K_D(\mathrm{nM}) \downarrow$, SA↓ and QED↑.

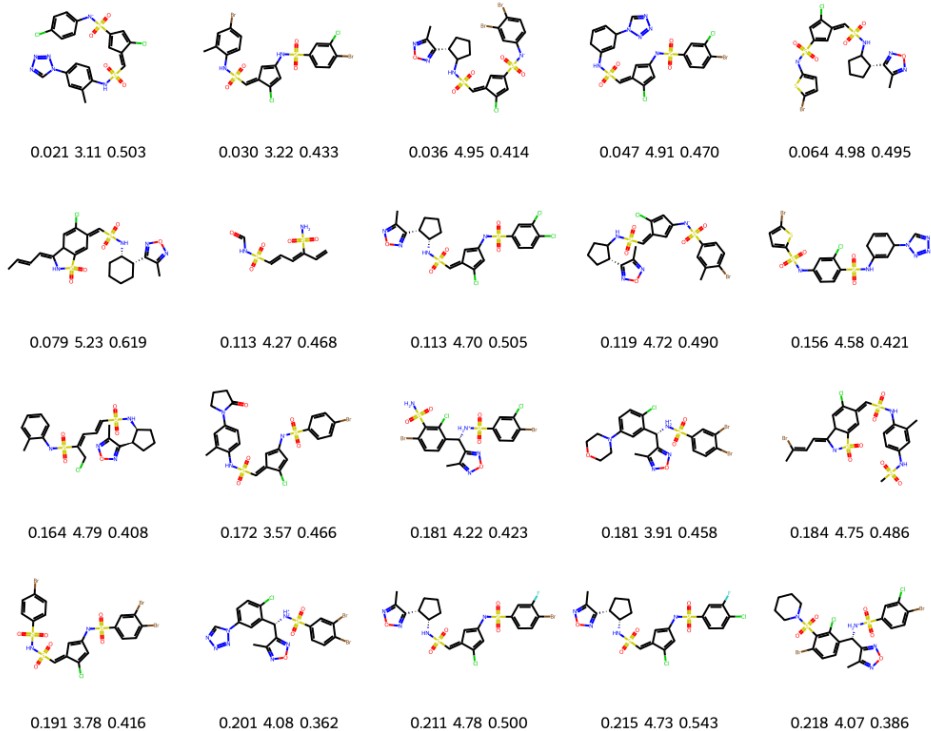

Figure 8: Molecules produced during the multi-objective optimization for PHGDH. The legends denote $K_D(\text{nM})\downarrow$, SA↓ and QED↑.

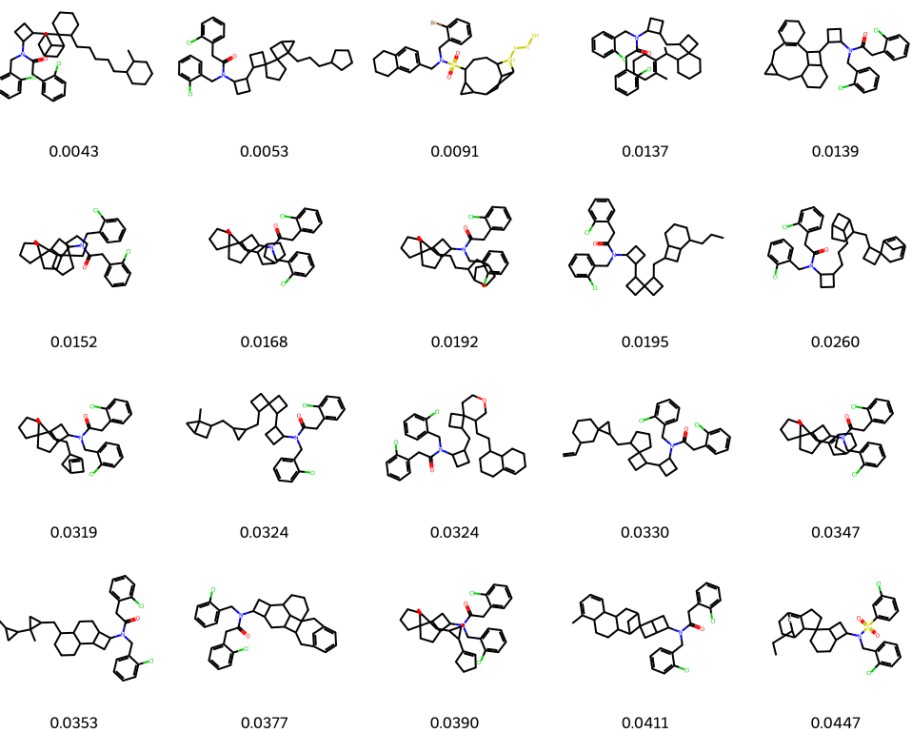

Figure 9: Molecules produced during the single-objective optimization for ESR1. The legends denote $\text{K}_\text{D}(nM)\downarrow$.

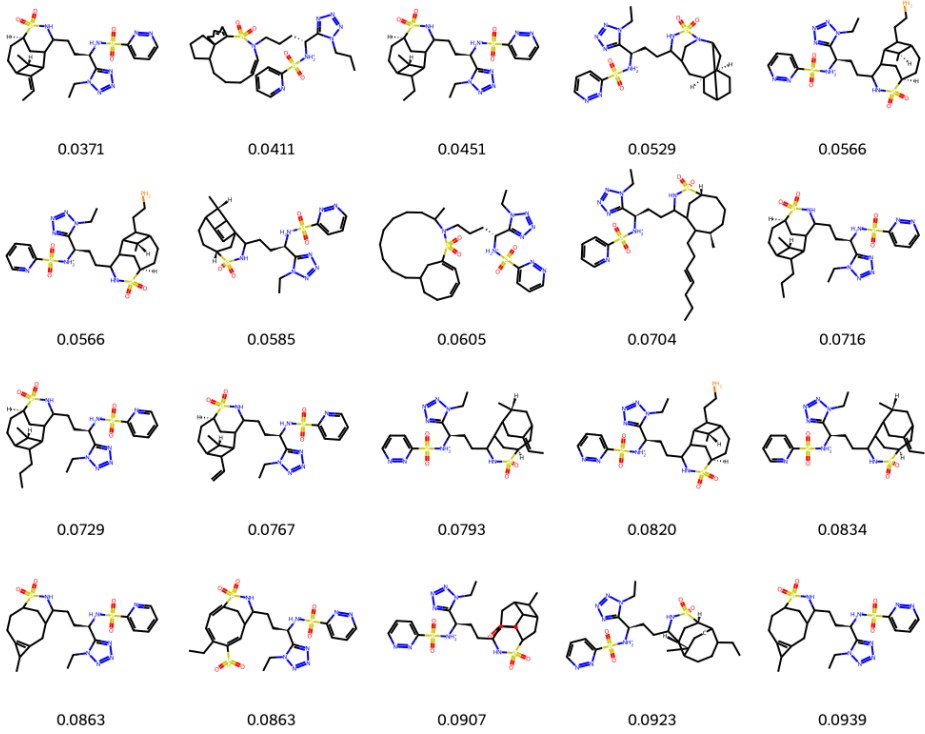

Figure 10: Molecules produced during the single-objective optimization for ACAA1. The legends denote $K_D(nM) \downarrow$.

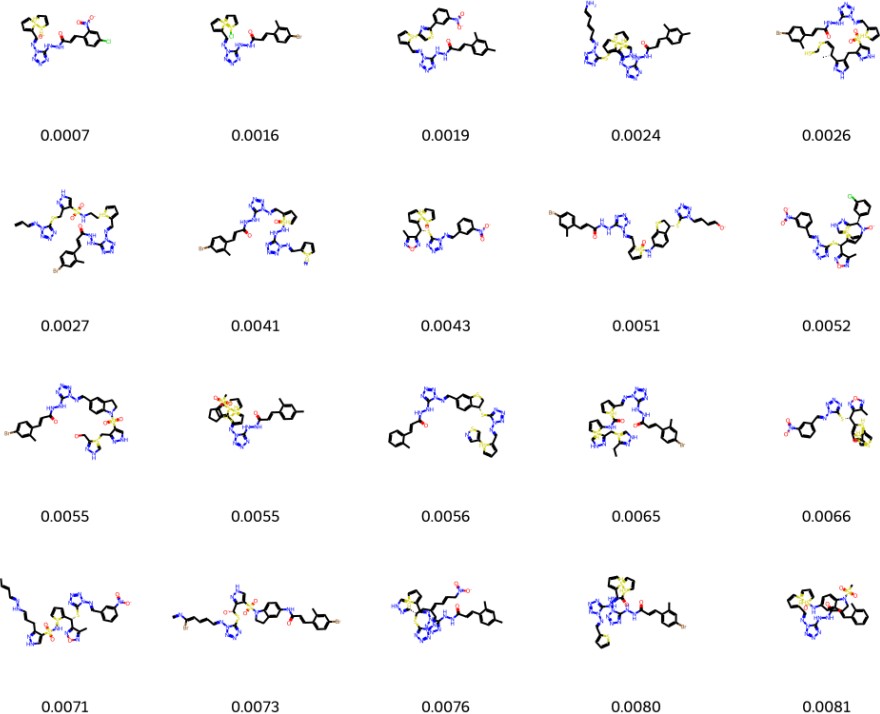

Figure 11: Molecules produced during the single-objective optimization for PHGDH. The legends denote $K_D(nM) \downarrow$.

### A.6.2 Structure-constrained Optimization

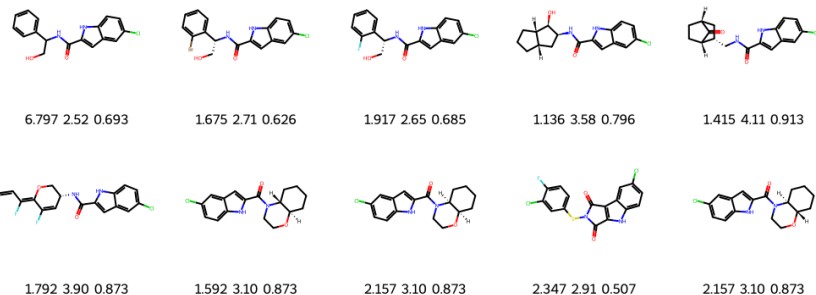

| | | | | |
|---|---|---|---|---|
| 6.797 2.52 0.693 | 1.675 2.71 0.626 | 1.917 2.65 0.685 | 1.136 3.58 0.796 | 1.415 4.11 0.913 |
| 1.792 3.90 0.873 | 1.592 3.10 0.873 | 2.157 3.10 0.873 | 2.347 2.91 0.507 | 2.157 3.10 0.873 |

Figure 12: Molecules produced during the structure-constrained optimization from C1 to C2 for PHGDH. The legends denote $K_D(\mu\mathrm{M}) \downarrow$, SA$\downarrow$ and QED$\uparrow$.

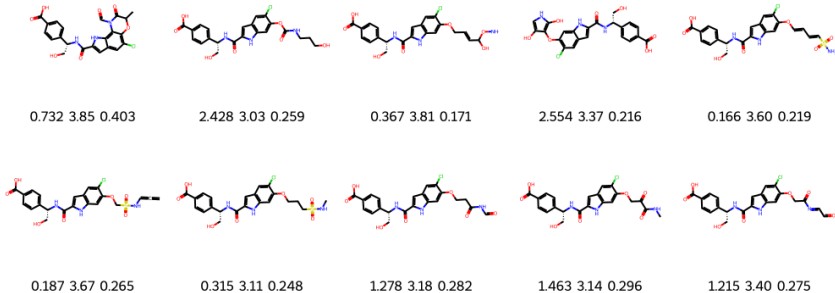

| | | | | |
|---|---|---|---|---|
| 0.732 3.85 0.403 | 2.428 3.03 0.259 | 0.367 3.81 0.171 | 2.554 3.37 0.216 | 0.166 3.60 0.219 |
| 0.187 3.67 0.265 | 0.315 3.11 0.248 | 1.278 3.18 0.282 | 1.463 3.14 0.296 | 1.215 3.40 0.275 |

Figure 13: Molecules produced during the structure-constrained optimization from C2 to C3 for PHGDH. The legends denote $K_D(\mu\mathrm{M}) \downarrow$, SA$\downarrow$ and QED$\uparrow$.

## A.7 Broader Impact

We introduce a novel generative model for jointly modeling molecule sequences and their target properties, which potentially leads to more capable and efficient molecule design algorithm. One potential negative impact could come from the misuse of the generative modeling in designing molecules or biological sequences for harmful purposes. Therefore, developing suitable safeguards and regulations will be crucial to mitigate potential negative impacts.

