# OpenReview forum: "Molecule Design by Latent Prompt Transformer"
_NeurIPS.cc/2024/Conference — NeurIPS 2024 spotlight_

### Official Review · Reviewer_vrpK · 2024-07-06

**Soundness:** 3
**Presentation:** 3
**Contribution:** 3
**Rating:** 7
**Confidence:** 3

**Summary:**

This paper presents LPT, a novel transformer model for conditional molecule sequence design and generation. LPT first generates latent vectors from a learnable prior distribution, then autoregressively generates molecule sequence taking the latent vector as prompt. Comprehensive experiments show that LPT achieves state-of-the-art performance on a variety of molecule, protein and DNA design benchmarks.

**Strengths:**

- Propose a novel transformer model LPT  and a novel framework to achieve property conditioned generation of molecule sequence.
- Strong and solid experiment results on multiple benchmark datasets.
- Good, clear and well-organized writing.

**Weaknesses:**

- As SMILES strings of molecules have grammars so not every SMILES string can be decoded to a real molecule. Authors are encouraged to report validity rate of the generated SMILES strings.
- What is the dimension/size of the latents in experiments? How will the dimension/size of the latents impact the performance? Authors are suggested to give a discussion or conduct ablation study experiments about this problem.

**Questions:**

See Weaknesses part.

**Limitations:**

Yes.

---

> ### Author Rebuttal · Authors · 2024-08-07
>
> We sincerely thank you for your thorough review and positive assessment of our paper. We appreciate your recognition of our novel model LPT, the framework for property-conditioned molecule sequence generation, and the strong experimental results across multiple benchmarks. We are pleased that you found our writing clear and well-organized.
> We would like to address the weaknesses and questions you raised:
>
> > W1: Validity rate of generated SMILES strings:
>
> Thank you for this important suggestion. In our molecule design experiments, we used the SELFIES representation (Krenn et al., 2020) instead of SMILES, as briefly mentioned in Section A.1.1. SELFIES is a 100% robust molecular string representation that ensures all generated strings correspond to valid molecules. As a result, we achieved 100% validity in all our molecule design experiments. We acknowledge that we should have made this point more explicit in the main text, and we will clearly state this in the revised paper.
>
> We agree that the validity checking of SMILES string generation is a good metric for model sanity check. Although not reported in the current version of the paper, we did perform this sanity check by evaluating SMILES generation with our model pretrained on ZINC. We achieved a validity rate of 0.999 for 10k generated molecules, which  validates our model's capability to generate valid molecules and the model can potentially capture the chemical rules. We will include this result in the revised paper.
>
> > W2: Dimension/size of latents and its impact on performance:
>
> Thank you for highlighting this important aspect. We conducted additional ablation studies on the impact of latent variable dimensionality in the challenging single-objective PHGDH experiment with the same number of oracle calls (50k):
> | Latent Variables  | 1st      | 2nd      |3rd      |Optimization Time (Hours)   |
> | -------- | ---------| -------- |-------- |-------- |
> |  2       |  0.10    |   0.14   | 0.21    | 8       |
> |  4       |  0.08    |   0.10   | 0.11    | 12      |
> |  8       |  0.07    |   0.09   | 0.12    | 15      |
>
> We observe a trade-off between performance and computational time. Increasing the number of latent variables improves results but increases optimization time. The performance gain diminishes from 4 to 8 variables, while computational cost continues to rise.
> Based on these findings, we chose 4 latent vectors for z (each size 256, total dimension 1024) in our main experiments, balancing performance and computational efficiency for practical applications. We shall include this discussion in our revised paper.
>
>
>
> Once again, we thank you for your constructive feedback. We are committed to improving our paper based on your suggestions.

---

> > ### Comment · Reviewer_vrpK · 2024-08-11
> > **Follow-up Response**
> >
> > I appreciate authors' efforts in rebuttal. All my concerns have been addressed so I will keep my rating.

---

> > > ### Author Response · Authors · 2024-08-11
> > > **Thank you!**
> > >
> > > Thank you for your positive feedback and thorough review. We’re glad all your concerns have been addressed. Have a nice day!

---

### Official Review · Reviewer_rWJi · 2024-07-12

**Soundness:** 2
**Presentation:** 3
**Contribution:** 2
**Rating:** 4
**Confidence:** 4

**Summary:**

The paper introduces an approach for molecule design, by leveraging recent advancements in conditional generative models for language and image generation. It contains three steps: (1) learnable prior distribution, (2) molecule generation model and (3) property prediction model. The experiments are comprehensive and convincing to me.

**Strengths:**

1. The task that the proposed method tries to solve is of great importance in a lot of real-world applications such drug design and protein design.
2. The formulation of multi-objective optimization task is quite novel and well defined.
3. The experiments are comprehensive and convincing to me.

**Weaknesses:**

1. Motivation to use the current framework is unclear to me. Why we use Langevin dynamics to sample latent variables rather than directly using the VAE framework to introduce the prior?
2. I think the dependence between molecules/properties cannot be secured as there is no reconstruction loss on molecules. As you may see in Eq. (5) and the equation under Eq. (6), the information when predicting y from z cannot be back-propagated to x. Then how to align x and y?
3. Minor:
(a) line 42: would suggest to write down the full name of MCMC before using its abbreviation for general audience.
(b) typo at equation (1): subscript of o^p_m(x).

**Questions:**

1. At line 29, I don't think existing methods decouple the training of the generative model from the property conditioned optimization, such as [1] and [2].
2. What's the difference between the proposed method and variational autoencoder (VAE)? I see both methods assume the prior on the latents, and there are works that generated molecules from VAE such as [1] and [2].
3. I noticed that the paper proposes Langevin dynamics to get z, but why not directly use an encoder to encode z from data like VAE does? MCMC-based method seems more computationally heavy to me.
4. At line 154, I'm still confused about how to sample z | y using Langevin dynamics. Especially how can we compute the gradient of the p(z_0|y)?

[1] Multi-objective Deep Data Generation with Correlated Property Control. NeurIPS 2022.
[2] Property controllable variational autoencoder via invertible mutual dependence. ICLR 2020.

**Limitations:**

Yes. Limitations are well discussed.

---

> ### Author Rebuttal · Authors · 2024-08-07
>
> Thank you for your detailed and constructive review. We appreciate the time and effort you’ve put into evaluating our work. We will address your comments and questions point by point:
>
> >W1: Motivation for framework:
>
> Thank you for your question. We kindly refer you to the **Global Response Point 1** for a detailed explanation of our motivation.
>
> >W2: Dependence between molecules/properties:
>
> Our approach ensures the dependence between molecules and properties through:
>
> 1.**Joint probability modeling**: We learn $p(x,y,z) = p(z)p(x|z)p(y|z)$, where $z = U_\alpha(z_0)$ and $z_0 \sim \mathcal{N}(0, I_d)$, tying molecules and properties through the shared latent space $z$.
>
> 2.**MLE learning of joint model** (Equation 5, Algorithm 1):
> a) Sample $z$ from $p(z|x,y)$ using Langevin dynamics:
> $z_{\tau+1} = z_\tau + s\nabla_z \log p(z_\tau|x,y) + \sqrt{2s}\epsilon_\tau$
> The gradient for sampling $z$ is (as in line 137):
> $\nabla_{z_0} \log p(z_0|x,y) = \nabla_{z_0} \log p_0(z_0) + \nabla_{z_0} \sum_{t=1}^T \log p(x^{(t)}|x^{(<t)},z) + \nabla_{z_0} \log p(y|z)$
> This ensures $z$ is sampled from regions consistent with both the observed molecule structure and properties, while the noise term $\epsilon_\tau$ enables exploration of the latent space.
> b) Update model parameters to maximize likelihood given sampled $z$ above.
>
> >W3: Minor issues:
>
> We’ll spell out MCMC on first use and correct the typo in Equation 1.
>
> >Q1: Decoupling of generative model and property optimization:
>
> Thank you for this important point. Our original statement was too broad. In our introduction, we primarily focused on the online molecule design. We will revise our statement to reflect this and makes sure to cite the paper you mentioned:
>
> "While some approaches to molecule design have treated generative modeling and property optimization separately, recent work such as [1] and [2] has shown promising integration of these aspects. Our approach builds upon these ideas, extending them in the context of online molecule design."
>
> >Q2: Comparsion of VAE and MCMC
>
> In the offline setting (Sec. 3.2), both MCMC-based and VAE-based algorithms are valid for approximating MLE learning. However, VAE-based methods with autoregressive decoding can suffer from posterior collapse without careful design. We conducted additional experiments on single-objective QED optimization with 25K molecule-property pairs in offline settings. We parameterized the encoder as Transformer models with the same layers but with full attention rather than causal attention. Performance degradation for our model was observed:
>
> | Model  | 1st      | 2nd      |3rd      |Top-50   |
> | -------- | ---------| -------- |-------- |-------- |
> |  LPT    |  0.948   |   0.947  | 0.947    | 0.940±0.003   |
> |  LPT-VAE     |  0.944   |   0.944  | 0.943    | 0.928±0.008   |
>
> In the online setting (Sec. 3.4), which is the main focus of our paper, MCMC-based LPT is more compatible with our needs. It can adapt to distribution shifts with a learned prior model and flexibly control exploration and exploitation to improve sample efficiency. MCMC-based optimization should be compared to other online optimization techniques such as Bayesian optimization (BO) with a pre-trained VAE. Our detailed comparisons on the PMO benchmark show that our methods outperform VAE-based methods (Table is shown in **Global Response Point 3**), even when strong BO techniques are used.
>
>
> >Q3: Computational efficiency of MCMC based methods.
>
> We clarify that with careful design, MCMC is not our computational bottleneck. We’ll include this discussion in the revised paper.
>
> In offline pretraining, we sample $z \sim p(z|x,y) \propto p(z)p(x|z)p(y|z)$, where $z = U_\alpha(z_0)$ with 15-step Langevin dynamics. In online learning, we use a persistent Markov chain that amortizes sampling across optimization steps, reducing steps to 2, thus enhancing real-world applicability.
>
> To address your concern comprehensively, we compare our method with strong VAE-based online optimization methods in PMO benchmark:
>
> **Comparison of model size and optimization time**: We compare optimization time after pretraining generative models on molecules with baselines in PMO bechmark with multiple-property objective (MPO) experiments. LPT shows comparable computation time to VAE-based Bayesion Optimization(BO) method.
>
> | Method           | Assemb.     | Model      | Pretrain | Model Size (M)  | Time (min)   |
> |------------------|-----------------|----------------|----------------|-----------------|-----------------|
> | VAE-BO SMILES    | SMILES      |  RNN & VAE       |   Y      |  17.9       | 17          |
> | VAE-BO SELFIES   | SELFIES     |  RNN & VAE       |   Y      |  18.7       | 21          |
> | LPT(Ours)              | SELFIES     |Transformer & MCMC |   Y      |  4.3        | 15          |
>
> >Q4: Sampling $p(z_0|y)$ using Langevin dynamics:
>
> This process is explained in Equation 7 of the main paper. The posterior distribution is given by: $z_0 \sim p_\theta(z_0|y) \propto p_0(z_0)p_\gamma(y|z = U_\alpha(z_0))$. To sample from this distribution using Langevin dynamics, we iterate:
> $z_0^{\tau+1} = z_0^\tau + s\nabla_{z_0} \log p_\theta(z_0|y) + \sqrt{2s}\epsilon^\tau$ where:
> $\nabla_{z_0} \log p_\theta(z_0|y) = \nabla_{z_0} \log p_0(z_0) + \nabla_{z_0} \log p_\gamma(y|z = U_\alpha(z_0))$
>
> Here, $p_0$ is a standard Gaussian distribution, so its gradient is straightforward to compute. The second term is also computable. For example, if we assume $p_\gamma(y|z)=\mathcal{N}(s_\gamma(z),\sigma^2)$ as in line 121, the second term becomes $1/\sigma^2(y-s_\gamma(z=U_\alpha(z_0)))\nabla_{z_0}s_\gamma(z=U_\alpha(z_0))$ using the chain rule. This can be implemented by pytorch autograd.
>
> Thank you for your valuable feedback on our submission. If you have any additional concerns, please let us know. We hope our responses have addressed your queries satisfactorily. If so, **we would appreciate your consideration in raising your rating of our submission**.

---

> > ### Author Response · Authors · 2024-08-11
> > **Have we addressed your concerns?**
> >
> > Dear Reviewer,
> >
> > As we approach the end of the discussion period, we wanted to check if we've adequately addressed your concerns. If you have any additional questions or if there are any points that require further clarification, please don't hesitate to let us know. We appreciate your time and valuable feedback.
> >
> > Thank you for your consideration.
> >
> > Authors

---

> > ### Comment · Reviewer_rWJi · 2024-08-12
> > **response**
> >
> > Thank author for the response which partially addressed my concern. Therefore I have increased my score. However, I'm still confused about several parts of the rebuttal.
> >
> > 1. I still cannot tell how Langevin dynamics can circumvents posterior collapse. I think it's just a method to sample from a distribution by the score function. It's pretty like VAE samples from Gaussian. The reviewer's rebuttal looks quite away from the insights of Langevin dynamics.
> >
> > 2. I think Eq. (4) only indicates that $x$ and $y$ are independent given $z$. From here we can say the alignment between $z$ and $x$, $z$ and $y$, but how to secure the dependence either between $z$ and joint $(x, y)$ or between $x$ and $y$? Simply, since in the end we aim to sample $x~p(x|y)$, how $p(x|y)$ be learned from paper's setting?

---

> ### Author Response · Authors · 2024-08-12
>
> Thank you for your continued feedback. We'd like to address two key points:
>
> >1. Posterior collapse in VAE.
>
> For the posterior collapse issue in VAEs, the early stages of learning can present a significant mismatch between the encoder and generator models. This mismatch causes the latent variable $z$, inferred by the encoder, to be highly inaccurate. Consequently, the autoregressive decoder model $p(x_t | x_{<t}, z)$ tends to disregard this imprecise latent variable. Instead, it models the input observation primarily using its own parameters and the input $x_{<t}$. This occurs because $p(x_t|x_{<t})$ is often strong enough to generate the molecule on its own, with minimal reliance on $z$. This behavior ultimately leads to posterior collapse.
>
> In contrast, Langevin dynamics approaches the problem differently. It consistently samples from the true posterior distribution of the latent variable, bypassing the need for a learned encoder and directly utilizing the autoregressive decoder. This method tends to be more accurate, especially in the initial stages of the learning process. As a result, the sampled $z$ consistently contributes to modeling the input observation through the decoder, mitigating the risk of posterior collapse.
>
> > 2. joint probability of $p(x,y,z)$
>
> The joint distribution of $(x, y, z)$ is $p(x, y, z) = p(z) p(x|z) p(y|z).$
> The joint distribution of $(x, y)$ is $p(x, y) = \int p(z) p(x|z) p(y|z) dz$, so that their dependency is captured by the sharing of $z$.
>
> $z$ plays the role of information bottleneck, i.e., $x$ is predicted from $y$ via $z$, i.e.,
>
> $$p(x|y) = \int p(x|z) p(z|y) dz.$$
>
> Given $(x, y)$, $z $ can be sampled from $p(z|x, y) \propto p(z) p(y|z) p(x|z)$ as a function of $z$ with $x$ and $y$ fixed.
>
> For molecule design, $z$ can be sampled from $p(z|y) \propto p(z) p(y|z)$ as a function of $z$ with $y$ fixed and we generate $x$ conditional on $z$, i.e. $x\sim p(x|z)$ as a form of ancestral sampling using derived $p(x|y)$ above.
>
> The sampling from both $p(z|x, y)$ and $p(z|y)$ can be accomplished by Langevin dynamics.
>
> We shall make it more explicit in the revised version of our paper.
>
> Again, thanks for your insightful comments. Please don't hesitate to let us know if you have any questions.

---

> > ### Author Response · Authors · 2024-08-13
> > **Have we addressed your concerns?**
> >
> > Dear Reviewer,
> >
> > Thank you again for your follow-up questions. With only a short time left in the discussion period, we wanted to ensure we've adequately addressed all of your concerns.
> >
> > If you have any additional questions or if any points require further clarification, please don't hesitate to let us know. We're eager to provide any necessary information to fully address your review.
> >
> > We sincerely appreciate your time and valuable insights and hope you will consider supporting and championing our work.
> >
> > Thank you for your thoughtful evaluation.
> >
> > Authors

---

### Official Review · Reviewer_jGf3 · 2024-07-12

**Soundness:** 3
**Presentation:** 3
**Contribution:** 3
**Rating:** 7
**Confidence:** 4

**Summary:**

The authors introduce a new conditional generative model for molecules. The model is called the Latent Prompt Transformer (LPT). A conditional model capable of generating new molecules with desired target properties is very useful in de-novo molecular design since we often want to design new molecules with some set of target properties. The authors initially train the LPT model on existing pairs of known molecules and property values. They then iteratively shift the distribution of the model towards regions with desired target properties. This results in a model capable of generating new molecules that are likely to have the desired target properties. In the experimental results section, the authors apply LPT to optimization tasks where the goal is to find new molecules that have high objective values, where the objective value here is some measurable desirable property of the molecule (i.e. the molecule’s binding affinity to some target of interest). Results show that the LPT model succeeds at these tasks, generating new molecules with higher binding affinity than baseline approaches. Additionally, the authors show that LPT can succeed at multi-objective optimization (generating new molecules with more than one desired target property). In particular, they show that LPT can generate new molecules that achieve high binding affinity, as well as high QED (Quantitative Estimate of Druglikeness) and minimal SA (synthetic accessibility) scores. In this case, results show that LPT is still able to generate molecules with higher binding affinity scores than baseline methods, while also having QED scores above 0.4 and SA scores below 5. Finally, the authors also show that LPT can be successfully applied to a new structure-constrained optimization task.

**Strengths:**

Originality: LPT is novel as it is (as far as I am aware) the first autoregressive conditional generative model designed specifically for the task of conditional molecule generation.
Quality: The paper is very well-written. Additionally, the figures and tables are all of good quality - they are both easy to parse and do a nice job of displaying results. Figures 1 and 2 do a very nice job of illustrating the author’s method. I especially like the use of Figure 2 to illustrate the distribution shift towards molecules with higher binding affinity.
Clarity: The paper is clear and easy to follow from start to finish. The figures and tables are clear and easy to read. The way the authors set up, trained, and applied the LPT model is clear.
Significance: De-novo molecular design is one of the most relevant/significant tasks in computational biology. In particular, finding molecules that bind to targets of interest is at the core of de-novo drug design. The paper shows that the author’s LPT model outperforms baseline approaches for relevant molecular design tasks such as finding new molecules with high binding affinity to targets of interest. This paper is therefore significant and of interest to the community.

**Weaknesses:**

Conditional generative models are not themselves novel as they have been successfully applied to a variety of text and image generation tasks. Much of the methodology in this paper involves taking existing methods in conditional generative modeling and applying them in a new domain: molecule generation. However, as far as I am aware this is the first time that one of these types of autoregressive conditional generative model has been designed for molecules, so this approach is novel from the perspective of methods for de-novo molecular design. I therefore think that this is a minor weakness and that this paper should be accepted.

Latent-space Bayesian optimization (LS-BO) approaches are mentioned in related work as an alternative approach, but not directly compared to in experiments. I actually don’t think that this direct comparison is strictly necessary for the paper to be accepted because LS-BO is an orthogonal approach and the authors do compare LPT to a good number of state-of-the-art generative modeling approaches. However, a direct comparison showing that LPT performs as well as state-of-the-art LS-BO methods would strengthen the paper.

**Questions:**

See Weaknesses section above for suggestions/points of discussion.

**Limitations:**

Yes.

---

> ### Author Rebuttal · Authors · 2024-08-07
>
> Thank you for your thoughtful and comprehensive review of our paper. We greatly appreciate your positive feedback on the originality, quality, clarity, and significance of our work. We're particularly grateful for your recognition of LPT's novelty and significance in de-novo molecule design.
>
> Regarding the weaknesses you identified:
>
> >W1. Novelty of conditional generative models:
>
> We  appreciate your insightful observation about the novelty of our approach. Indeed, while conditional generative models have been applied in other domains, we believe our work makes a significant contribution by being the first to combine latent variable modeling with autoregressive molecule generation specifically for molecule design. This combination leverages the expressive power of Transformer-based autoregressive modeling for molecule generation and the efficiency of low-dimensional latent space sampling and design. This synergy is particularly advantageous in addressing the complexities of molecular structures and navigating the vast chemical space. Furthermore, it enhances our online learning algorithm, improving both design outcomes and sample efficiency. This integrated approach facilitates more effective exploration and optimization in molecular design.
>
> >W2. Comparison with Latent-space Bayesian Optimization (LS-BO):
>
> We appreciate this suggestion and have added additional experiments comparing LPT with prominent BO-based optimization methods, which will be included in the revised version of our paper. We report the multi-property objectives (MPO) experiments under a limited oracle budget of 10K queries on the practical molecular optimization (PMO) benchmark [Gao et al., 2022]. We include the strongest BO-based baseline Gaussian process BO (GP-BO) [Tripp et al., 2021], LS-BO baselines including VAE-BO with both trained with SMILES [Gómez-Bombarelli et al., 2018] and SELFIES strings, and LSTM HC SMILES, which is the best generative molecule design method in PMO and is the same as reported in Table 5 of our main paper. Note that GP-BO optimizes the GP acquisition function with graph genetic algorithm in an inner loop.
>
> | Method           | Amlodipine      | Fexofenadine   | Osimertinib    | Perindopril     | Ranolazine      | Zaleplon        | Sum   |
> |------------------|-----------------|----------------|----------------|-----------------|-----------------|-----------------|-------|
> | LSTM HC SMILES   | 0.593±0.016     | **0.725±0.003**    | **0.796±0.002**    | 0.489±0.007     | 0.714±0.008     | 0.206±0.006     | 3.523 |
> | GP-BO              | 0.583±0.044     | 0.722±0.005    | 0.787±0.006    | 0.493±0.011     | **0.735±0.013**     | 0.221±0.072     | 3.541 |
> | VAE-BO SMILES              | 0.533±0.009     | 0.671±0.003    | 0.771±0.002    | 0.442±0.004     | 0.457±0.012     | 0.039±0.012     | 2.913 |
> | VAE-BO SELFIES              | 0.516±0.005     | 0.670±0.004    | 0.765±0.002    | 0.429±0.003     | 0.452±0.025     | 0.206±0.015     | 3.038 |
> | LPT              | **0.608±0.005**     | 0.714±0.003    | 0.784±0.011    | **0.511±0.002**     | 0.682±0.007     | **0.245±0.003**     | **3.544** |
>
> As shown, LPT performs comparably to the strong GP-BO baseline and LSTM HC SMILES, while significantly outperforming LS-BO (VAE based) methods across multiple tasks. LPT achieves the highest overall sum score, demonstrating its robust performance across diverse molecular optimization objectives. Since BO-based baselines often require careful parameter tuning, we directly report the numbers from the PMO benchmark that is carefully tuned. In future work, we plan to conduct more thorough comparisons in binding affinity experiments to further validate LPT's performance against these baselines.
>
> References:
>
> [1] W. Gao et al. Sample Efficiency Matters: A Benchmark for Practical Molecular Optimization. NeurIPS, 2022.
>
> [2] A. Tripp et al. A fresh look at de novo molecular design benchmarks. NeurIPS 2021 AI for Science Workshop, 2021.
>
> [3] R. Gómez-Bombarelli et al. Automatic chemical design using a data-driven continuous representation of molecules. ACS central science, 2018.
>
> We believe these additions further solidify our work's contribution to de-novo molecular design. Thank you again for your valuable feedback and recommendation for acceptance. We look forward to the opportunity to present a strengthened version of our work that incorporates your valuable suggestions.

---

> > ### Author Response · Authors · 2024-08-12
> > **Thank you for reviewing our work.**
> >
> > Dear Reviewer,
> >
> > As we approach the end of the discussion period, we wanted to ensure we've adequately addressed all of your concerns.
> >
> > If you have any additional questions or if any points require further clarification, please don't hesitate to let us know. We're eager to provide any necessary information to fully address your review.
> >
> > We sincerely appreciate your time and valuable insights. We hope you will consider supporting and championing our work.
> >
> > Thank you for your thoughtful evaluation.
> >
> > Best,
> >
> > Authors

---

> > > ### Comment · Reviewer_jGf3 · 2024-08-12
> > > **Rebuttal Acknowledgement**
> > >
> > > I would like to thank authors for their response and addressing the points I raised in my review. I am happy to keep my assessment of their work the same.

---

> > > > ### Author Response · Authors · 2024-08-13
> > > > **Thank you!**
> > > >
> > > > Thank you for your positive feedback and thorough review. We're glad all your concerns have been addressed. We really appreciate your continued support.
> > > >
> > > > Best,
> > > >
> > > > Authors

---

### Official Review · Reviewer_D3n3 · 2024-07-14

**Soundness:** 3
**Presentation:** 2
**Contribution:** 3
**Rating:** 4
**Confidence:** 4

**Summary:**

This work proposes a novel molecular optimization framework Latent Prompt Transformer (LPT)，modeling latent distribution, molecule sequences and properties distributions conditioned on the latent distribution. They uses MCMC in MLE training and conditional generation. Additionally, they propose an online learning algorithm to extrapolate to feasible regions supporting the desired property. The framework demonstrates promising results across various molecular design tasks.

**Strengths:**

1. The online learning approach aligns well with real-world scenarios.
2. The experiments are comprehensive, and the introduction of a new task for conditional generation and design trajectory comparison is expressive.
3. The paper discusses and makes efforts to alleviate the bottleneck of computational efficiency. It also improves sample efficiency by reweighting, making the method more feasible.

**Weaknesses:**

- The clarity of the paper could be improved in several aspects:
  1. Adding an illustration or diagram of the training and optimization process would make the methodology easier to understand.
  2. Some formulas lack derivations and explanations, as raised in Question 1.
- I am concerned that using MCMC sampling for training and inference may impact the model's practicality. Could you please compare the training and sampling speed with the baselines and discuss this issue in the limitations section?
- Minor Issues:
1. Typo error: "2rd" in Table 1.
2. Using "Kd" values might lead readers to mistakenly think they are wet lab results, while in paper they represent docking scores.
3. In Algorithm 2, 'step (c) Update LPT on synthetic dataset using **Algorithm 2** using MLE.' Should it be Algorithm 1 here?

**Questions:**

1. How is the transition from equation 4 to equation 5 derived? It would be beneficial to provide a detailed derivation for this step, as it is not immediately apparent.
2. When formulating p(y∣z), how do you choose σ\sigmaσ and how does it affect performance? Since properties can be uniquely determined by the molecule, i.e., p(y∣x) is deterministic, this suggests that in the probability model shown in Fig. 1 (Left), p(y∣z) should be very deterministic. However, this creates difficulties for sampling p_\theta(z_0|y) in eq.7, because when \sigma is small, \nabla\log p_\theta(z_0|y) is very small for most z_0, leading to very slow MCMC convergence, especially with poor initialization.
3. How robust is your model to the quality of the Oracle?

**Limitations:**

Yes.

---

> ### Author Rebuttal · Authors · 2024-08-07
>
> Thank you for your thorough review. We appreciate your recognition of our work's alignment with real-world scenarios, comprehensive experiments, and efficiency improvements. We'll address your comments and questions point by point.
>
> > W1. Clarity improvements:
>
> We'll add a diagram illustrating the training and optimization process in the revised version and provide more detailed derivations (See Q1).
> > W2. MCMC sampling practicality:
>
> Thank you for raising this important point. We hope to clarify that with careful design, MCMC is not our computational bottleneck. We'll include this discussion in the revised paper.
>
> In offline pretraining, we sample $z \sim p(z|x,y) \propto p(z)p(x|z)p(y|z)$, where $z = U_\alpha(z_0)$ with 15-step Langevin dynamics. In online learning or optimization, we use a persistent Markov chain that amortize the sampling across all optimization steps, reducing steps to 2. This speedup focuses on the optimization phase for real-world applicability.
>
> To address your concern comprehensively, we add:
>
> 1. **Comparison of model size and optimization time**: We compare optimization time after pretraining generative models on molecules. LPT shows comparable computation time to Bayesian optimization (BO) methods with much lower model size for LPT. This is for multiple-property objective (MPO) experiments in Practical Molecular Optimization (PMO) from Sec. 5.4.
>
>   | Method           | Assemb.     | Model      | Pretrain | Model Size (M)  | Time (min)   |
>   |------------------|-----------------|----------------|----------------|-----------------|-----------------|
>   | LSTM HC SMILES   | SMILES      |  RNN       |   Y      |  98.9       | 3           |
>   | VAE-BO SMILES    | SMILES      |  RNN & VAE       |   Y      |  17.9       | 17          |
>   | VAE-BO SELFIES   | SELFIES     |  RNN & VAE       |   Y      |  18.7       | 21          |
>   | LPT(Ours)              | SELFIES     |Transformer & MCMC |   Y      |  4.3        | 15          |
>
> 2. **Breakdown of major computational costs for a single batch of 256 samples**:
>
>   | Computational Costs(s)   | Offline pretraining      | Online optimization     |
>   | --------------------    | --------------| -------------- |
>   | Posterior Sampling of $z$          |  5.147        |   0.294        |
>   | Molecule Generation     |  0.021        |   0.993        |
>   | Property Prediction     |  0.039        |   -        |
>
> Thanks for the question. We shall consider improving the offline model learning speed in the future work and discuss this explicitly in the limiations.
>
> >W3. Minor issues:
>    - We'll correct the typo "2rd" to "2nd" in Table 1.
>    - Good point about "Kd" values. We'll clarify these are computational predictions, not wet lab results.
>    - You're correct, it should be Algorithm 1 in Algorithm 2 step \(c\). We'll fix this error.
>
> > Q1. Derivation from equation 4 to 5:
>
> We'll provide a detailed derivation in the revision. Please refer to **Global Response Point 1**.
>
>
> >Q2. Formulating $p(y|z)$ and $\sigma$ choice:
>
> Thank you for this insightful question. You're correct that $p(y|x)$ is deterministic by nature. However, introducing a small error term $\sigma$ in generative models is a common and necessary practice, aligning with various regression models and latent space optimization techniques.
> - In probabilistic regression models, including Gaussian Process Regression, $y = f(x) + \epsilon$, where $\epsilon \sim N(0, \sigma^2)$. Here, $\sigma$ represents observation noise or model uncertainty as in Gómez-Bombarelli et al. (2018).
> - In Variational Autoencoders (VAE) for continuous variables, the property encoder typically outputs both a mean $\mu$ and a standard deviation $\sigma$ for each dimension, modeling $p(z|y)$ as $N(\mu, \sigma^2)$ such as Jiang et al.(NeurIPS 2020).
> - Latent Space Bayesian Optimization methods like LSBO (Tripp et al., NeurIPS 2020) use similar noise terms when fitting its Gaussian Process surrogate model.
>
> In our case, modeling $p(y|z)$ with a small $\sigma$ serves several purposes:
> - Account for potential inaccuracies in the oracle function or property prediction.
> - Provide smoothness to the optimization landscape, potentially aiding in convergence.
> - Allow for a degree of uncertainty in the latent space, which can be beneficial for exploration (in online learning LPT).
>
> You raise an excellent point about the challenges in sampling $p_\theta(z_0|y)$ when $\sigma$ is small. One possible solution is to anneal $\sigma$, by starting from a big value and gradually reducing it.
>
> References:
>
> [1] Gómez-Bombarelli et al. Automatic chemical design using a data-driven continuous representation of molecules. (ACS central science 2018)
>
> [2] Jiang et al. Multi-objective Deep Data Generation with Correlated Property Control. (NeurIPS 2020)
>
> [3] Tripp et al. "Sample-Efficient Optimization in the Latent Space of Deep Generative Models via Weighted Retraining" (NeurIPS 2020)
>
> >Q3. Model robustness to Oracle quality:
>
> We've added experiments varying Oracle noise levels to assess robustness on single-objective QED optimization tasks under the budget of 25K. We use noised oracles as $y_{noise}=y_{true}+e$, where $e\sim N(0, \sigma^2)$, varying $\sigma$ as a percentage of the property range. The results show our model is robust to the noised oracles.
>
> | Oracle Noise  | 1st      | 2nd      |3rd      |Top-50   |
> | -------- | ---------| -------- |-------- |-------- |
> |  w/o     |  0.948   |   0.947  | 0.947    | 0.940±0.003   |
> |  1%      |  0.947   |   0.947  | 0.946    | 0.939±0.004   |
> |  5%      |  0.946   |   0.946  | 0.944    | 0.936±0.005   |
> |  10%     |  0.945   |   0.945  | 0.942    | 0.932±0.006   |
>
> Thank you for these insightful comments. They will help improve the paper's clarity and completeness. We hope our responses have addressed your queries satisfactorily. If so, **we would appreciate your consideration in raising your rating of our submission**.

---

> > ### Author Response · Authors · 2024-08-11
> > **Have we addressed your concerns?**
> >
> > Dear Reviewer,
> >
> > As the discussion period nears its end, we wanted to check if we've adequately addressed your concerns. If you have any additional questions or if there are any points that require further clarification, please don't hesitate to let us know. We appreciate your time and valuable feedback.
> >
> > Best,
> >
> > Authors

---

> ### Author Response · Authors · 2024-08-13
> **Have we addressed your concerns?**
>
> Dear Reviewer,
>
> With only a short time left in the discussion period, we wanted to ensure we've fully addressed your concerns. If you have any additional questions or require further clarification on any points, please don't hesitate to reach out. We greatly appreciate your time and valuable feedback.
>
> Best regards,
>
> Authors

---

### Author Rebuttal · Authors · 2024-08-07

Dear Reviewers,

Thank you for your insightful and constructive comments on our submission. We have added derivation of Eqn. 5 for completeness and clarified our motivation for using MCMC-based methods in online molecule design.

1. **Derivation of Equation 5**

$$
\begin{align*}
   \nabla_\theta \log p_\theta(x, y)
   &= \frac{\nabla_\theta p_\theta(x, y)}{p_\theta(x, y)} \\\\
   &= \frac{1}{p_\theta(x, y)} \int \nabla_\theta p_\theta(x, y, z_0)  dz_0 \\\\
   &= \int \frac{p_\theta(x, y, z_0)}{p_\theta(x, y)} \nabla_\theta \log p_\theta(x, y, z_0) dz_0 \\\\
   &= \int p_\theta(z_0 | x, y) \nabla_\theta \log p_\theta(x, y, z_0) dz_0 \\\\
   &= \mathbb{E_{p_\theta(z_0|x, y)}} \left[ \nabla_\theta \log p_\theta(x, y, z_0) \right] \\\\
   &= \mathbb{E_{p_\theta(z_0|x, y)}} \left[ \nabla_\theta \log p_\beta(x|U_\alpha(z_0)) + \nabla_\theta \log p_\gamma(y|U_\alpha(z_0)) + \nabla_\theta \log p_0(z_0) \right] \\\\
   &= \mathbb{E_{p_\theta(z_0|x, y)}} \left[ \nabla_\theta \log p_\beta(x|U_\alpha(z_0)) + \nabla_\theta \log p_\gamma(y|U_\alpha(z_0)) \right].
\end{align*}
$$




2. **Motivation of MCMC-based learning and optimization**
Our choice of Langevin dynamics for online molecule design was driven by several key advantages:

- **Avoiding posterior collapse**: Langevin dynamics circumvents the issue of posterior collapse, a significant challenge in training VAEs with autoregressive decoders for sequential data like molecules. In VAEs, the strong autoregressive decoder can often reconstruct the data by relying solely on the one-step-ahead ground truth, ignoring the latent codes entirely. This leads to a trivial local optimum where the posterior collapses to the prior, carrying no useful information [1]. In contrast, our method samples from the posterior distribution of z, iteratively refining the latent prompts and increasing their likelihood given the observed data. This iterative refinement ensures that the latent space remains informative throughout training, avoiding the collapse problem inherent in VAEs [2].

   [1] Fu, Hao, et al. “Cyclical annealing schedule: A simple approach to mitigating kl vanishing.” NAACL (2019).

   [2] Pang, Bo, et al. “Generative text modeling through short run inference.” EACL (2021).
- **Adaptability to distribution shifting with a learnable prior**: Our method incorporates a learnable prior, crucial for adapting to scenarios where desired property values lie outside the initially learned distribution. This adaptability is essential in practical online molecule design, where we aim to generate molecules with properties not present in the initial training data. Unlike VAEs with fixed priors, our learnable prior adjusts to new target regions in the property space with a small number of online data samples. This is quantitatively demonstrated in our ablation study (Appendix A1.2, Table 7), which shows significant performance degradation when using a fixed Gaussian prior.
- **Flexible exploration-exploitation trade-off**: Our use of Langevin dynamics for property-conditioned generation allows nuanced control over the exploration-exploitation trade-off during test-time. As shown in Equation 8 and the ablation study in Table 8, we can adjust the guidance weight in the sampling process to balance between exploring new molecular structures and exploiting known high-performing regions of the latent space. This flexibility is crucial for improving sample efficiency in practical molecule design. There are existing works that leverage pretrained VAEs and other optimization techniques (such as Bayesian Optimization) to adapt VAEs in online scenarios.
- **Unfied framework**:  MCMC sampling allows us to develop both offline and online learning algorithms within a unified framework as LPT.
- **Handling multi-modal posteriors**: The posterior distribution of latent variables given a desired property may be multi-modal. Langevin dynamics is better equipped to handle such multi-modal distributions compared to the typically unimodal approximate posteriors used in VAEs.

3. **Comaprsion with VAE-based optimization methods.**
We provide additional comparsion with VAE-based methods on the practical molecular optimization(PMO), where multi-property objectives are optimized under a limited budget of 10K. The numbers are taken from PMO benchmark.

| Method           | Amlodipine      | Fexofenadine   | Osimertinib    | Perindopril     | Ranolazine      | Zaleplon        | Sum   |
|------------------|-----------------|----------------|----------------|-----------------|-----------------|-----------------|-------|
| VAE-BO SMILES              | 0.533±0.009     | 0.671±0.003    | 0.771±0.002    | 0.442±0.004     | 0.457±0.012     | 0.039±0.012     | 2.913 |
| JT-VAE BO Fragments              | 0.519±0.009     | 0.667±0.003    | 0.775±0.002    | 0.430±0.004     | 0.508±0.012     | 0.046±0.012     | 2.945 |
| VAE-BO SELFIES              | 0.516±0.005     | 0.670±0.004    | 0.765±0.002    | 0.429±0.003     | 0.452±0.025     | 0.206±0.015     | 3.038 |
| LPT SELFIES             | **0.608±0.005**     | **0.714±0.003**    | **0.784±0.011**    | **0.511±0.002**     | **0.682±0.007**     | **0.245±0.003**     | **3.544** |

---

### Decision · Program_Chairs · 2024-09-25

**Decision:**

Accept (spotlight)

**Comment:**

The reviewers have appreciated the novelty of the approach, although some of it stemming from the fact that the approach was applied to molecules for the first time; they also liked the fact that the experimental work was comprehensive. The reviewers had concerns about the computational weight of the MCMC sampling, but those were alleviated already by the proposed approach and additional comparisons to SOTAs were performed during the rebuttal period to show practical efficiency of the method. The authors contributed a number of additional results including comparisons to additional methods during the rebuttal. Overall the paper is a solid contribution and of interest to the NeurIPS community.